# Improved Sample Complexity for Incremental Autonomous Exploration in MDPs

**Jean Tarbouriech**
Facebook AI Research Paris & Inria Lille
jean.tarbouriech@gmail.com

**Matteo Pirotta**
Facebook AI Research Paris
pirotta@fb.com

**Michal Valko**
DeepMind Paris
valkom@deepmind.com

**Alessandro Lazaric**
Facebook AI Research Paris
lazaric@fb.com

## Abstract

We investigate the exploration of an unknown environment when no reward function is provided. Building on the incremental exploration setting introduced by Lim and Auer [1], we define the objective of learning the set of $\varepsilon$-optimal goal-conditioned policies attaining all states that are incrementally reachable within $L$ steps (in expectation) from a reference state $s_0$. In this paper, we introduce a novel model-based approach that interleaves discovering new states from $s_0$ and improving the accuracy of a model estimate that is used to compute goal-conditioned policies to reach newly discovered states. The resulting algorithm, DisCo, achieves a sample complexity scaling as $\widetilde{O}(L^5 S_{L+\varepsilon} \Gamma_{L+\varepsilon} A \, \varepsilon^{-2})$, where $A$ is the number of actions, $S_{L+\varepsilon}$ is the number of states that are incrementally reachable from $s_0$ in $L + \varepsilon$ steps, and $\Gamma_{L+\varepsilon}$ is the branching factor of the dynamics over such states. This improves over the algorithm proposed in [1] in both $\varepsilon$ and $L$ at the cost of an extra $\Gamma_{L+\varepsilon}$ factor, which is small in most environments of interest. Furthermore, DisCo is the first algorithm that can return an $\varepsilon/c_{\min}$-optimal policy for any cost-sensitive shortest-path problem defined on the $L$-reachable states with minimum cost $c_{\min}$. Finally, we report preliminary empirical results confirming our theoretical findings.

## 1 Introduction

In cases where the reward signal is not informative enough — e.g., too sparse, time-varying or even absent — a reinforcement learning (RL) agent needs to explore the environment driven by objectives other than reward maximization, see [e.g., 2, 3, 4, 5, 6]. This can be performed by designing intrinsic rewards to drive the learning process, for instance via state visitation counts [7, 8], novelty or prediction errors [9, 10, 11]. Other recent methods perform information-theoretic skill discovery to learn a set of diverse and task-agnostic behaviors [12, 13, 14]. Alternatively, goal-conditioned policies learned by carefully designing the sequence of goals during the learning process are often used to solve sparse reward problems [15] and a variety of goal-reaching tasks [16, 17, 18, 19].

While the approaches reviewed above effectively leverage deep RL techniques and are able to achieve impressive results in complex domains (e.g., Montezuma's Revenge [15] or real-world robotic manipulation tasks [19]), they often lack substantial theoretical understanding and guarantees. Recently, some *unsupervised RL* objectives were analyzed rigorously. Some of them quantify how well the agent visits the states under a sought-after frequency, e.g., to induce a maximally entropic state distribution [20, 21, 22, 23]. While such strategies provably mimic their desired behavior via a Frank-Wolfe algorithmic scheme, they may not learn how to effectively reach any state of the environment and thus may not be sufficient to efficiently solve downstream tasks. Another relevant take is the reward-free RL paradigm of [24]: following its exploration phase, the agent is able to

compute a near-optimal policy for any reward function at test time. While this framework yields strong end-to-end guarantees, it is limited to the finite-horizon setting and the agent is thus unable to tackle tasks beyond finite-horizon, e.g., goal-conditioned tasks.

In this paper, we build on and refine the setting of incremental exploration of [1]: the agent starts at an initial state $s_0$ in an unknown, possibly large environment, and it is provided with a RESET action to restart at $s_0$. At a high level, in this setting the agent should explore the environment and stop when it has identified the *tasks* within its *reach* and learned to *master* each of them sufficiently well. More specifically, the objective of the agent is to learn a goal-conditioned policy for *any* state that can be reached from $s_0$ within $L$ steps in expectation; such a state is said to be $L$-controllable. Lim and Auer [1] address this setting with the UcbExplore method for which they bound the number of exploration steps that are required to identify in an incremental way all $L$-controllable states (i.e., the algorithm needs to define a suitable stopping condition) and to return a set of policies that are able to reach each of them in *at most* $L + \varepsilon$ steps. A key aspect of UcbExplore is to first focus on simple states (i.e., states that can be reached within a few steps), learn policies to efficiently reach them, and leverage them to identify and tackle states that are increasingly more difficult to reach. This approach aims to avoid wasting exploration in the attempt of reaching states that are further than $L$ steps from $s_0$ or that are too difficult to reach given the limited knowledge available at earlier stages of the exploration process. Our main contributions are:

- We strengthen the objective of incremental exploration and require the agent to learn $\varepsilon$-optimal goal-conditioned policies for any $L$-controllable state. Formally, let $V^\star(s)$ be the length of the shortest path from $s_0$ to $s$, then the agent needs to learn a policy to navigate from $s_0$ to $s$ in at most $V^\star(s) + \varepsilon$ steps, while in [1] any policy reaching $s$ in *at most* $L + \varepsilon$ steps is acceptable.
- We design DisCo, a novel algorithm for incremental exploration. DisCo relies on an estimate of the transition model to compute goal-conditioned policies to the states observed so far and then use those policies to improve the accuracy of the model and incrementally discover new states.
- We derive a sample complexity bound for DisCo scaling as[1] $\widetilde{O}(L^5 S_{L+\varepsilon} \Gamma_{L+\varepsilon} A \varepsilon^{-2})$, where $A$ is the number of actions, $S_{L+\varepsilon}$ is the number of states that are *incrementally* controllable from $s_0$ in $L + \varepsilon$ steps, and $\Gamma_{L+\varepsilon}$ is the branching factor of the dynamics over such incrementally controllable states. Not only is this sample complexity obtained for a more challenging objective than UcbExplore, but it also improves in both $\varepsilon$ and $L$ at the cost of an extra $\Gamma_{L+\varepsilon}$ factor, which is small in most environments of interest.
- Leveraging the model-based nature of DisCo, we can also readily compute an $\varepsilon/c_{\min}$-optimal policy for *any* cost-sensitive shortest-path problem defined on the $L$-controllable states with minimum cost $c_{\min}$. This result serves as a goal-conditioned counterpart to the reward-free exploration framework defined by Jin et al. [24] for the finite-horizon setting.

## 2 Incremental Exploration to Discover and Control

In this section we expand [1], with a more challenging objective for autonomous exploration.

### 2.1 $L$-Controllable States

We consider a *reward-free* Markov decision process [25, Sect. 8.3] $M := \langle \mathcal{S}, \mathcal{A}, p, s_0 \rangle$. We assume a finite action space $\mathcal{A}$ with $A = |\mathcal{A}|$ actions, and a finite, possibly large state space $\mathcal{S}$ for which an upper bound $S$ on its cardinality is known, i.e., $|\mathcal{S}| \leq S$.[2] Each state-action pair $(s, a) \in \mathcal{S} \times \mathcal{A}$ is characterized by an unknown transition probability distribution $p(\cdot|s, a)$ over next states. We denote by $\Gamma_{\mathcal{S}'} := \max_{s \in \mathcal{S}', a} \|\{p(s'|s, a)\}_{s' \in \mathcal{S}'}\|_0$ the largest branching factor of the dynamics over states in any subset $\mathcal{S}' \subseteq \mathcal{S}$. The environment has no extrinsic reward, and $s_0 \in \mathcal{S}$ is a designated initial state.

A deterministic stationary policy $\pi : \mathcal{S} \to \mathcal{A}$ is a mapping between states to actions and we denote by $\Pi$ the set of all possible policies. Since in environments with arbitrary dynamics the learner may get stuck in a state without being able to return to $s_0$, we introduce the following assumption.[3]

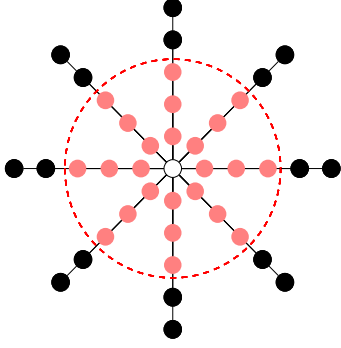
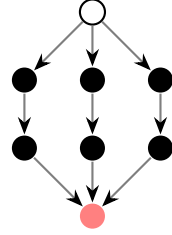

Figure 1: Two environments where the starting state $s_0$ is in white. *Left:* Each transition between states is deterministic and depicted with an edge. *Right:* Each transition from $s_0$ to the first layer is *equiprobable* and the transitions in the successive layers are deterministic. If we set $L = 3$, then the states belonging to $\mathcal{S}_L$ are colored in red. As the right figure illustrates, $L$-controllability is not necessarily linked to a notion of distance between states and an $L$-controllable state may be achieved by traversing states that are not $L$-controllable themselves.

**Assumption 1.** *The action space contains a* RESET *action s.t. $p(s_0|s, \text{RESET}) = 1$ for any $s \in \mathcal{S}$.*

We make explicit the states where a policy $\pi$ takes action RESET in the following definition.

**Definition 1** (Policy restricted on a subset). *For any $\mathcal{S}' \subseteq \mathcal{S}$, a policy $\pi$ is restricted on $\mathcal{S}'$ if $\pi(s) = \text{RESET}$ for any $s \notin \mathcal{S}'$. We denote by $\Pi(\mathcal{S}')$ the set of policies restricted on $\mathcal{S}'$.*

We measure the performance of a policy in navigating the MDP as follows.

**Definition 2.** *For any policy $\pi$ and a pair of states $(s, s') \in \mathcal{S}^2$, let $\tau_\pi(s \to s')$ be the (random) number of steps it takes to reach $s'$ starting from $s$ when executing policy $\pi$, i.e., $\tau_\pi(s \to s') := \inf\{t \geq 0 : s_{t+1} = s' \,|\, s_1 = s, \pi\}$. We also set $v_\pi(s \to s') := \mathbb{E}[\tau_\pi(s \to s')]$ as the expected traveling time, which corresponds to the value function of policy $\pi$ in a stochastic shortest-path setting (SSP, [26, Sect. 3]) with initial state $s$, goal state $s'$ and unit cost function. Note that we have $v_\pi(s \to s') = +\infty$ when the policy $\pi$ does not reach $s'$ from $s$ with probability 1. Furthermore, for any subset $\mathcal{S}' \subseteq \mathcal{S}$ and any state $s$, we denote by*

$$V^\star_{\mathcal{S}'}(s_0 \to s) := \min_{\pi \in \Pi(\mathcal{S}')} v_\pi(s_0 \to s),$$

*the length of the shortest path to $s$, restricted to policies resetting to $s_0$ from any state outside $\mathcal{S}'$.*

The objective of the learning agent is to *control efficiently* the environment in the *vicinity* of $s_0$. We say that a state $s$ is controlled if the agent can reliably navigate to it from $s_0$, that is, there exists an effective *goal-conditioned policy* — i.e., a *shortest-path policy* — from $s_0$ to $s$.

**Definition 3** ($L$-controllable states). *Given a reference state $s_0$, we say that a state $s$ is $L$-controllable if there exists a policy $\pi$ such that $v_\pi(s_0 \to s) \leq L$. The set of $L$-controllable states is then*

$$\mathcal{S}_L := \{s \in \mathcal{S} : \min_{\pi \in \Pi} v_\pi(s_0 \to s) \leq L\}. \tag{1}$$

We illustrate the concept of controllable states in Fig. 1 for $L = 3$. Interestingly, in the right figure, the black states are not $L$-controllable. In fact, there is no policy that can directly choose which one of the black states to reach. On the other hand, the red state, despite being in some sense *further* from $s_0$ than the black states, *does* belong to $\mathcal{S}_L$. In general, there is a crucial difference between the existence of a *random* realization where a state $s$ is reached from $s_0$ in less than $L$ steps (i.e., black states) and the notion of $L$-*controllability*, which means that there exists a policy that consistently reaches the state in a number of steps less or equal than $L$ on average (i.e., red state). This explains the choice of the term *controllable* over *reachable*, since a state $s$ is often said to be reachable if there is a policy $\pi$ with a non-zero probability to eventually reach it, which is a weaker requirement.

Unfortunately, Lim and Auer [1] showed that in order to discover all the states in $\mathcal{S}_L$, the learner may require a number of exploration steps that is *exponential* in $L$ or $|\mathcal{S}_L|$. Intuitively, this negative result is due to the fact that the minimum in Eq. 1 is over the set of all possible policies, including those that may traverse states that are not in $\mathcal{S}_L$.[4] Hence, we similarly constrain the learner to focus on the set of *incrementally controllable* states.

**Definition 4** (Incrementally controllable states $\mathcal{S}_L^\to$). *Let $\prec$ be some partial order on $\mathcal{S}$. The set $\mathcal{S}_L^\prec$ of states controllable in $L$ steps w.r.t. $\prec$ is defined inductively as follows. The initial state $s_0$*

belongs to $\mathcal{S}_L^{\prec}$ by definition and if there exists a policy $\pi$ restricted on $\{s' \in \mathcal{S}_L^{\prec} : s' \prec s\}$ with $v_\pi(s_0 \to s) \leq L$, then $s \in \mathcal{S}_L^{\prec}$. The set $\mathcal{S}_L^{\to}$ of incrementally $L$-controllable states is defined as $\mathcal{S}_L^{\to} := \cup_\prec \mathcal{S}_L^{\prec}$, where the union is over all possible partial orders.

By way of illustration, in Fig. 1 for $L = 3$, it holds that $\mathcal{S}_L^{\to} = \mathcal{S}_L$ in the left figure, whereas $\mathcal{S}_L^{\to} = \{s_0\} \neq \mathcal{S}_L$ in the right figure. Indeed, while the red state is $L$-controllable, it requires traversing the black states, which are not $L$-controllable.

## 2.2 AX Objectives

We are now ready to formalize two alternative objectives for *Autonomous eXploration* (AX) in MDPs.

**Definition 5** (AX sample complexity). *Fix any length $L \geq 1$, error threshold $\varepsilon > 0$ and confidence level $\delta \in (0, 1)$. The sample complexities $\mathcal{C}_{\mathrm{AX}_L}(\mathfrak{A}, L, \varepsilon, \delta)$ and $\mathcal{C}_{\mathrm{AX}^\star}(\mathfrak{A}, L, \varepsilon, \delta)$ are defined as the number of time steps required by a learning algorithm $\mathfrak{A}$ to identify a set $\mathcal{K} \supseteq \mathcal{S}_L^{\to}$ such that with probability at least $1 - \delta$, it has learned a set of policies $\{\pi_s\}_{s \in \mathcal{K}}$ that respectively verifies the following* AX *requirement*

$(\mathrm{AX_L}) \quad \forall s \in \mathcal{K}, v_{\pi_s}(s_0 \to s) \leq L + \varepsilon,$

$(\mathrm{AX}^\star) \quad \forall s \in \mathcal{K}, v_{\pi_s}(s_0 \to s) \leq V_{\mathcal{S}_L^{\to}}^\star(s_0 \to s) + \varepsilon.$

Designing agents satisfying the objectives defined above introduces critical difficulties w.r.t. standard goal-directed learning in RL. First, the agent has to find accurate policies for a set of goals (i.e., all incrementally $L$-controllable states) and not just for one specific goal. On top of this, the set of desired goals itself (i.e., the set $\mathcal{S}_L^{\to}$) is *unknown* in advance and has to be estimated online. Specifically, $\mathrm{AX_L}$ is the original objective introduced in [1] and it requires the agent to discover all the incrementally $L$-controllable states as fast as possible.[5] At the end of the learning process, for each state $s \in \mathcal{S}_L^{\to}$ the agent should return a policy that can reach $s$ from $s_0$ in at most $L$ steps (in expectation). Unfortunately, this may correspond to a rather poor performance in practice. Consider a state $s \in \mathcal{S}_L^{\to}$ such that $V_{\mathcal{S}_L^{\to}}^\star(s_0 \to s) \ll L$, i.e., the shortest path between $s_0$ to $s$ following policies restricted on $\mathcal{S}_L^{\to}$ is much smaller than $L$. Satisfying $\mathrm{AX_L}$ only guarantees that a policy reaching $s$ in $L$ steps is found. On the other hand, objective $\mathrm{AX}^\star$ is more demanding, as it requires learning a near-optimal shortest-path policy for each state in $\mathcal{S}_L^{\to}$. Since $V_{\mathcal{S}_L^{\to}}^\star(s_0 \to s) \leq L$ and the gap between the two quantities may be arbitrarily large, especially for states close to $s_0$ and far from the fringe of $\mathcal{S}_L^{\to}$, $\mathrm{AX}^\star$ is a significantly tighter objective than $\mathrm{AX_L}$ and it is thus preferable in practice.

We say that an exploration algorithm solves the AX problem if its sample complexity $\mathcal{C}_{\mathrm{AX}}(\mathfrak{A}, L, \varepsilon, \delta)$ in Def. 5 is polynomial in $|\mathcal{K}|$, $A$, $L$, $\varepsilon^{-1}$ and $\log(S)$. Notice that requiring a logarithmic dependency on the size of $\mathcal{S}$ is crucial but nontrivial, since the overall state space may be large and we do not want the agent to waste time trying to reach states that are not $L$-controllable. The dependency on the (algorithmic-dependent and random) set $\mathcal{K}$ can be always replaced using the upper bound $|\mathcal{K}| \leq |\mathcal{S}_{L+\varepsilon}^{\to}|$, which is implied with high probability by both $\mathrm{AX_L}$ and $\mathrm{AX}^\star$ conditions. Finally, notice that the error threshold $\varepsilon > 0$ has a two-fold impact on the performance of the algorithm. First, $\varepsilon$ defines the largest set $\mathcal{S}_{L+\varepsilon}^{\to}$ that could be returned by the algorithm: the larger $\varepsilon$, the bigger the set. Second, as $\varepsilon$ increases, the quality (in terms of controllability and navigational precision) of the output policies worsens w.r.t. the shortest-path policy restricted on $\mathcal{S}_L^{\to}$.

## 3 The `DisCo` Algorithm

The algorithm `DisCo` — for `Discover and Control` — is detailed in Alg. 1. It maintains a set $\mathcal{K}$ of "controllable" states and a set $\mathcal{U}$ of states that are considered "uncontrollable" *so far*. A state $s$ is tagged as controllable when a policy to reach $s$ in at most $L + \varepsilon$ steps (in expectation from $s_0$) has been found with high confidence, and we denote by $\pi_s$ such policy. The states in $\mathcal{U}$ are states that have been discovered as potential members of $\mathcal{S}_L^{\to}$, but the algorithm has yet to produce a policy to control any of them in less than $L + \varepsilon$ steps. The algorithm stores an estimate of the transition model and it proceeds through rounds, which are indexed by $k$ and incremented whenever a state in $\mathcal{U}$ gets transferred to the set $\mathcal{K}$, i.e., when the transition model reaches a level of accuracy sufficient

**Algorithm 1:** Algorithm `DisCo`
---

**Input:** Actions $\mathcal{A}$, initial state $s_0$, confidence parameter $\delta \in (0,1)$, error threshold $\varepsilon > 0$, $L \geq 1$ and (possibly adaptive) allocation function $\phi : \mathcal{P}(\mathcal{S}) \to \mathbb{N}$ (where $\mathcal{P}(\mathcal{S})$ denotes the power set of $\mathcal{S}$).

**1** Initialize $k := 0$, $\mathcal{K}_0 := \{s_0\}$, $\mathcal{U}_0 := \{\}$ and a restricted policy $\pi_{s_0} \in \Pi(\mathcal{K}_0)$.

**2** Set $\varepsilon := \min\{\varepsilon, 1\}$ and `continue := True`.

**3 while** `continue` **do**

**4**     Set $k \mathrel{+}= 1$. //new round
      // ① Sample collection on $\mathcal{K}$

**5**     For each $(s,a) \in \mathcal{K}_k \times \mathcal{A}$, execute policy $\pi_s$ until the total number of visits $N_k(s,a)$ to $(s,a)$ satisfies
      $N_k(s,a) \geq n_k := \phi(\mathcal{K}_k)$. For each $(s,a) \in \mathcal{K}_k \times \mathcal{A}$, add $s' \sim p(\cdot|s,a)$ to $\mathcal{U}_k$ if $s' \notin \mathcal{K}_k$.
      // ② Restriction of candidate states $\mathcal{U}$

**6**     Compute transitions $\widehat{p}_k(s'|s,a)$ and $\mathcal{W}_k := \left\{ s' \in \mathcal{U}_k : \exists (s,a) \in \mathcal{K}_k \times \mathcal{A}, \widehat{p}_k(s'|s,a) \geq \frac{1-\varepsilon/2}{L} \right\}$.

**7**     **if** $\mathcal{W}_k$ is empty **then**

**8**       $\lfloor$ Set `continue := False`. //condition STOP1

**9**     **else**
      // ③ Computation of the optimistic policies on $\mathcal{K}$

**10**       **for** each state $s' \in \mathcal{W}_k$ **do**

**11**         $\lfloor$ Compute $(\widetilde{u}_{s'}, \widetilde{\pi}_{s'}) := \mathrm{OVI_{SSP}}(\mathcal{K}_k, \mathcal{A}, s', N_k, \frac{\varepsilon}{6L})$, see Alg. 3 in App. D.1.

**12**       Let $s^\dagger := \arg\min_{s \in \mathcal{W}_k} \widetilde{u}_s(s_0)$ and $\widetilde{u}^\dagger := \widetilde{u}_{s^\dagger}(s_0)$.

**13**       **if** $\widetilde{u}^\dagger > L$ **then**

**14**         $\lfloor$ Set `continue := False`. //condition STOP2

**15**       **else**
        // ④ State transfer from $\mathcal{U}$ to $\mathcal{K}$

**16**         $\lfloor$ Set $\mathcal{K}_{k+1} := \mathcal{K}_k \cup \{s^\dagger\}$, $\mathcal{U}_{k+1} := \mathcal{U}_k \setminus \{s^\dagger\}$ and $\pi_{s^\dagger} := \widetilde{\pi}_{s^\dagger}$.

    // ⑤ Policy consolidation: computation on the final set $\mathcal{K}$

**17** Set $K := k$.

**18 for** each state $s \in \mathcal{K}_K$ **do**

**19**     $\lfloor$ Compute $(\widetilde{u}_s, \widetilde{\pi}_s) := \mathrm{OVI_{SSP}}(\mathcal{K}_K, \mathcal{A}, s, N_K, \frac{\varepsilon}{6L})$.

**20 Output:** the states $s$ in $\mathcal{K}_K$ and their corresponding policy $\pi_s := \widetilde{\pi}_s$.

---

to compute a policy to control one of the states encountered before. We denote by $\mathcal{K}_k$ (resp. $\mathcal{U}_k$) the set of controllable (resp. uncontrollable) states at the beginning of round $k$. `DisCo` stops at a round $K$ when it can confidently claim that all the remaining states outside of $\mathcal{K}_K$ cannot be $L$-controllable.

At each round, the algorithm uses all samples observed so far to build an estimate of the transition model denoted by $\widehat{p}(s'|s,a) = N(s,a,s')/N(s,a)$, where $N(s,a)$ and $N(s,a,s')$ are counters for state-action and state-action-next state visitations. Each round is divided into two phases. The first is a *sample collection* phase. At the beginning of round $k$, the agent collects additional samples until $n_k := \phi(\mathcal{K}_k)$ samples are available at each state-action pair in $\mathcal{K}_k \times \mathcal{A}$ (step ①). A key challenge lies in the careful (and adaptive) choice of the allocation function $\phi$, which we report in the statement of Thm. 1 (see Eq. 19 in App. D.4 for its exact definition). Importantly, the incremental construction of $\mathcal{K}_k$ entails that sampling at each state $s \in \mathcal{K}_k$ can be done efficiently. In fact, for all $s \in \mathcal{K}_k$ the agent has already confidently learned a policy $\pi_s$ to reach $s$ in at most $L + \varepsilon$ steps on average (see how such policy is computed in the second phase). The generation of transitions $(s,a,s')$ for $(s,a) \in \mathcal{K}_k \times \mathcal{A}$ achieves two objectives at once. First, it serves as a discovery step, since all observed next states $s'$ not in $\mathcal{U}_k$ are added to it — in particular this guarantees sufficient exploration at the fringe (or border) of the set $\mathcal{K}_k$. Second, it improves the accuracy of the model $p$ in the states in $\mathcal{K}_k$, which is essential in computing near-optimal policies and thus fulfilling the AX$^\star$ condition.

The second phase does not require interacting with the environment and it focuses on the *computation of optimistic policies*. The agent begins by significantly restricting the set of candidate states in each round to alleviate the computational complexity of the algorithm. Namely, among all the states in $\mathcal{U}_k$, it discards those that do not have a high probability of belonging to $\mathcal{S}_L^{\rightarrow}$ by considering a restricted set $\mathcal{W}_k \subseteq \mathcal{U}_k$ (step ②). In fact, if the estimated probability $\widehat{p}_k$ of reaching a state $s \in \mathcal{U}_k$ from any of the controllable states in $\mathcal{K}_k$ is lower than $(1 - \varepsilon/2)/L$, then no shortest-path policy restricted on $\mathcal{K}_k$ could get to $s$ from $s_0$ in less than $L + \varepsilon$ steps on average. Then for each state $s'$ in $\mathcal{W}_k$, `DisCo` computes an optimistic policy restricted on $\mathcal{K}_k$ to reach $s'$. Formally, for any candidate state $s' \in \mathcal{W}_k$, we define the induced stochastic shortest path (SSP) MDP $M'_k$ with goal state $s'$ as follows.

**Definition 6.** *We define the SSP-MDP $M'_k := \langle \mathcal{S}, \mathcal{A}'_k(\cdot), c'_k, p'_k \rangle$ with goal state $s'$, where the action space is such that $\mathcal{A}'_k(s) = \mathcal{A}$ for all $s \in \mathcal{K}_k$ and $\mathcal{A}'_k(s) = \{\text{RESET}\}$ otherwise (i.e., we focus on policies restricted on $\mathcal{K}_k$). The cost function is such that for all $a \in \mathcal{A}$, $c'_k(s', a) = 0$, and for any $s \neq s'$, $c'_k(s, a) = 1$. The transition model is $p'_k(s'|s', a) = 1$ and $p'_k(\cdot|s, a) = p(\cdot|s, a)$ otherwise.[6]*

The solution of $M'_k$ is the shortest-path policy from $s_0$ to $s'$ restricted on $\mathcal{K}_k$. Since $p'_k$ is unknown, DisCo cannot compute the exact solution of $M'_k$, but instead, it executes optimistic value iteration (OVI$_{\text{SSP}}$) for SSP [27, 28] to obtain a value function $\widetilde{u}_{s'}$ and its associated greedy policy $\widetilde{\pi}_{s'}$ restricted on $\mathcal{K}_k$ (see App. D.1 for more details).

The agent then chooses a candidate goal state $s^\dagger$ for which the value $\widetilde{u}^\dagger := \widetilde{u}_{s^\dagger}(s_0)$ is the smallest. This step can be interpreted as selecting the optimistically most promising new state to control. Two cases are possible. If $\widetilde{u}^\dagger \leq L$, then $s^\dagger$ is added to $\mathcal{K}_k$ (step ④), since the accuracy of the model estimate on the state-action space $\mathcal{K}_k \times \mathcal{A}$ guarantees that the policy $\widetilde{\pi}_{s^\dagger}$ is able to reach the state $s^\dagger$ in less than $L + \varepsilon$ steps in expectation with high probability (i.e., $s^\dagger$ is incrementally $(L + \varepsilon)$-controllable). Otherwise, we can guarantee that $\mathcal{S}_L^{\rightarrow} \subseteq \mathcal{K}_k$ with high probability. In the latter case, the algorithm terminates and, using the current estimates of the model, it recomputes an optimistic shortest-path policy $\pi_s$ restricted on the final set $\mathcal{K}_K$ for each state $s \in \mathcal{K}_K$ (step ⑤). This policy consolidation step is essential to identify near-optimal policies restricted on the final set $\mathcal{K}_K$ (and thus on $\mathcal{S}_L^{\rightarrow}$): indeed the expansion of the set of the so far controllable states may alter and refine the optimal goal-reaching policies restricted on it (see App. A).

**Computational Complexity.** Note that algorithmically, we do not need to define $M'_k$ (Def. 6) over the whole state space $\mathcal{S}$ as we can limit it to $\mathcal{K}_k \cup \{s'\}$, i.e., the candidate state $s'$ and the set $\mathcal{K}_k$ of so far controllable states. As shown in Thm. 1, this set can be significantly smaller than $\mathcal{S}$. In particular this implies that the computational complexity of the value iteration algorithm used to compute the optimistic policies is independent from $S$ (see App. D.9 for more details).

## 4  Sample Complexity Analysis of DisCo

We now present our main result: a sample complexity guarantee for DisCo for the AX$^\star$ objective, which directly implies that AX$_L$ is also satisfied.

**Theorem 1.** *There exists an absolute constant $\alpha > 0$ such that for any $L \geq 1$, $\varepsilon \in (0, 1]$, and $\delta \in (0, 1)$, if we set the allocation function $\phi$ as*

$$\phi : \mathcal{X} \to \alpha \cdot \left( \frac{L^4 \widehat{\Theta}(\mathcal{X})}{\varepsilon^2} \log^2 \left( \frac{LSA}{\varepsilon \delta} \right) + \frac{L^2 |\mathcal{X}|}{\varepsilon} \log \left( \frac{LSA}{\varepsilon \delta} \right) \right), \quad (2)$$

*with $\widehat{\Theta}(\mathcal{X}) := \max_{(s,a) \in \mathcal{X} \times \mathcal{A}} \left( \sum_{s' \in \mathcal{X}} \sqrt{\widehat{p}(s'|s, a)(1 - \widehat{p}(s'|s, a))} \right)^2$, then the algorithm DisCo (Alg. 1) satisfies the following sample complexity bound for AX$^\star$*

$$\mathcal{C}_{\text{AX}^\star}(\text{DisCo}, L, \varepsilon, \delta) = \widetilde{O}\left( \frac{L^5 \Gamma_{L+\varepsilon} S_{L+\varepsilon} A}{\varepsilon^2} + \frac{L^3 S_{L+\varepsilon}^2 A}{\varepsilon} \right), \quad (3)$$

*where $S_{L+\varepsilon} := |\mathcal{S}_{L+\varepsilon}^{\rightarrow}|$ and*

$$\Gamma_{L+\varepsilon} := \max_{(s,a) \in \mathcal{S}_{L+\varepsilon}^{\rightarrow} \times \mathcal{A}} \|\{p(s'|s, a)\}_{s' \in \mathcal{S}_{L+\varepsilon}^{\rightarrow}}\|_0 \leq S_{L+\varepsilon}$$

*is the maximal support of the transition probabilities $p(\cdot|s, a)$ restricted to the set $\mathcal{S}_{L+\varepsilon}^{\rightarrow}$.*

Given the definition of AX$^\star$, Thm. 1 implies that DisCo **1)** terminates after $\mathcal{C}_{\text{AX}^\star}(\text{DisCo}, L, \varepsilon, \delta)$ time steps, **2)** discovers a set of states $\mathcal{K} \supseteq \mathcal{S}_L^{\rightarrow}$ with $|\mathcal{K}| \leq S_{L+\varepsilon}$, **3)** and for each $s \in \mathcal{K}$ outputs a policy $\pi_s$ which is $\varepsilon$-optimal w.r.t. policies restricted on $\mathcal{S}_L^{\rightarrow}$, i.e., $v_{\pi_s}(s_0 \to s) \leq V_{\mathcal{S}_L^{\rightarrow}}^{\star}(s_0 \to s) + \varepsilon$. Note that Eq. 3 displays only a *logarithmic* dependency on $S$, the total number of states. This property on the sample complexity of DisCo, along with its $S$-independent computational complexity, is significant when the state space $\mathcal{S}$ grows large w.r.t. the unknown set of interest $\mathcal{S}_L^{\rightarrow}$.

### 4.1 Proof Sketch of Theorem 1

While the complete proof is reported in App. D, we now provide the main intuition behind the result.

**State Transfer from $\mathcal{U}$ to $\mathcal{K}$ (step ④).** Let us focus on a round $k$ and a state $s^\dagger \in \mathcal{U}_k$ that gets added to $\mathcal{K}_k$. For clarity we remove in the notation the round $k$, goal state $s^\dagger$ and starting state $s_0$. We denote by $v$ and $\widetilde{v}$ the value functions of the candidate policy $\widetilde{\pi}$ in the true and optimistic model respectively, and by $\widetilde{u}$ the quantity w.r.t. which $\widetilde{\pi}$ is optimistically greedy. We aim to prove that $s^\dagger \in \mathcal{S}_{L+\varepsilon}^{\rightarrow}$ (with high probability). The main chain of inequalities underpinning the argument is

$$v \le |v - \widetilde{v}| + \widetilde{v} \overset{(a)}{\le} \frac{\varepsilon}{2} + \widetilde{v} \overset{(b)}{\le} \frac{\varepsilon}{2} + \widetilde{u} + \frac{\varepsilon}{2} \overset{(c)}{\le} L + \varepsilon, \tag{4}$$

where (c) is guaranteed by algorithmic construction and (b) stems from the chosen level of value iteration accuracy. Inequality (a) has the flavor of a simulation lemma for SSP, by relating the shortest-path value function of a same policy between two models (the true one and the optimistic one). Importantly, when restricted to $\mathcal{K}$ these two models are close in virtue of the algorithmic design which enforces the collection of a minimum amount of samples at each state-action pair of $\mathcal{K} \times \mathcal{A}$, denoted by $n$. Specifically, we obtain that

$$|v - \widetilde{v}| = \widetilde{O}\Big(\sqrt{\frac{L^4 \Gamma_\mathcal{K}}{n}} + \frac{L^2 |\mathcal{K}|}{n}\Big), \qquad \text{with} \quad \Gamma_\mathcal{K} := \max_{(s,a) \in \mathcal{K} \times \mathcal{A}} \|\{p(s'|s,a)\}_{s' \in \mathcal{K}}\|_0 \le |\mathcal{K}|.$$

Note that $\Gamma_\mathcal{K}$ is the branching factor restricted to the set $\mathcal{K}$. Our choice of $n$ (given in Eq. 2) is then dictated to upper bound the above quantity by $\varepsilon/2$ in order to satisfy inequality (a). Let us point out that, interestingly yet unfortunately, the structure of the problem does not appear to allow for technical variance-aware improvements seeking to lower the value of $n$ prescribed above (indeed the AX framework requires to analytically encompass the uncontrollable states $\mathcal{U}$ into a single meta state with higher transitional uncertainty, see App. D for details).

**Termination of the Algorithm.** Since $\mathcal{S}_L^{\rightarrow}$ is *unknown*, we have to ensure that none of the states in $\mathcal{S}_L^{\rightarrow}$ are "missed". As such, we prove that with overwhelming probability, we have $\mathcal{S}_L^{\rightarrow} \subseteq \mathcal{K}_K$ when the algorithm terminates at a round denoted by $K$. There remains to justify the final near-optimal guarantee w.r.t. the set of policies $\Pi(\mathcal{S}_L^{\rightarrow})$. Leveraging that step ⑤ recomputes the policies $(\pi_s)_{s \in \mathcal{K}_K}$ on the final set $\mathcal{K}_K$, we establish the following chain of inequalities

$$v \le |v - \widetilde{v}| + \widetilde{v} \overset{(a)}{\le} \frac{\varepsilon}{2} + \widetilde{v} \overset{(b)}{\le} \frac{\varepsilon}{2} + \widetilde{u} + \frac{\varepsilon}{2} \overset{(c)}{\le} V_{\mathcal{K}_K}^\star + \varepsilon \overset{(d)}{\le} V_{\mathcal{S}_L^{\rightarrow}}^\star + \varepsilon, \tag{5}$$

where (a) and (b) are as in Eq. 4, (c) leverages optimism and (d) stems from the inclusion $\mathcal{S}_L^{\rightarrow} \subseteq \mathcal{K}_K$.

**Sample Complexity Bound.** The choice of allocation function $\phi$ in Eq. 2 bounds $n_K$ which is the total number of samples required at each state-action pair in $\mathcal{K}_K \times \mathcal{A}$. We then compute a high-probability bound $\psi$ on the time steps needed to collect a given sample, and show that it scales as $\widetilde{O}(L)$. Since the sample complexity is solely induced by the sample collection phase (step ①), it can be bounded by the quantity $\psi\, n_K |\mathcal{K}_K| A$. Putting everything together yields the bound of Thm. 1.

### 4.2 Comparison with `UcbExplore` [1]

We start recalling the critical distinction that `DisCo` succeeds in tackling problem $AX^\star$, while `UcbExplore` [1] fails to do so (see App. A for details on the AX objectives). Nonetheless, in the following we show that even if we restrict our attention to $AX_L$, for which `UcbExplore` is designed, `DisCo` yields a better sample complexity in most of the cases. From [1], `UcbExplore` verifies[7]

$$\mathcal{C}_{AX_L}(\texttt{UcbExplore}, L, \varepsilon, \delta) = \widetilde{O}\Big(\frac{L^6 S_{L+\varepsilon} A}{\varepsilon^3}\Big). \tag{6}$$

Eq. 6 shows that the sample complexity of `UcbExplore` is linear in $S_{L+\varepsilon}$, while for `DisCo` the dependency is somewhat worse. In the main-order term $\widetilde{O}(1/\varepsilon^2)$ of Eq. 3, the bound depends linearly on $S_{L+\varepsilon}$ but also grows with the branching factor $\Gamma_{L+\varepsilon}$, which is not the "global" branching factor

but denotes the number of possible next states in $\mathcal{S}_{L+\varepsilon}^{\rightarrow}$ starting from $\mathcal{S}_{L+\varepsilon}^{\rightarrow}$. While in general we only have $\Gamma_{L+\varepsilon} \leq S_{L+\varepsilon}$, in many practical domains (e.g., robotics, user modeling), each state can only transition to a small number of states, i.e., we often have $\Gamma_{L+\varepsilon} = O(1)$ as long as the dynamics is not too "chaotic". While DisCo does suffer from a quadratic dependency on $S_{L+\varepsilon}$ in the second term of order $\widetilde{O}(1/\varepsilon)$, we notice that for any $S_{L+\varepsilon} \leq L^3 \varepsilon^{-2}$ the bound of DisCo is still preferable. Furthermore, since for $\varepsilon \to 0$, $S_{L+\varepsilon}$ tends to $S_L$, the condition is always verified for small enough $\varepsilon$.

Compared to DisCo, the sample complexity of UcbExplore is worse in both $\varepsilon$ and $L$. As stressed in Sect. 2.2, the better dependency on $\varepsilon$ both improves the quality of the output goal-reaching policies as well as reduces the number of incrementally $(L + \varepsilon)$-controllable states returned by the algorithm. It is interesting to investigate why the bound of [1] (Eq. 6) inherits a $\widetilde{O}(\varepsilon^{-3})$ dependency. As reviewed in App. E, UcbExplore alternates between two phases of state discovery and policy evaluation. The optimistic policies computed by UcbExplore solve a *finite-horizon problem* (with horizon set to $H_{\mathrm{UCB}}$). However, minimizing the expected time to reach a target state is intrinsically an SSP problem, which is exactly what DisCo leverages. By computing policies that solve a finite-horizon problem (note that UcbExplore resets every $H_{\mathrm{UCB}}$ time steps), [1] sets the horizon to $H_{\mathrm{UCB}} := \lceil L + L^2 \varepsilon^{-1} \rceil$, which leads to a policy-evaluation phase with sample complexity scaling as $\widetilde{O}(H_{\mathrm{UCB}} \varepsilon^{-2}) = \widetilde{O}(\varepsilon^{-3})$. Since the rollout budget of $\widetilde{O}(\varepsilon^{-3})$ is hard-coded into the algorithm, the dependency on $\varepsilon$ of UcbExplore's sample complexity cannot be improved by a more refined analysis; instead a different algorithmic approach is required such as the one employed by DisCo.

### 4.3  Goal-Free Cost-Free Exploration on $\mathcal{S}_L^{\rightarrow}$ with DisCo

A compelling advantage of DisCo is that it achieves an accurate estimation of the environment's dynamics restricted to the unknown subset of interest $\mathcal{S}_L^{\rightarrow}$. In contrast to UcbExplore which needs to restart its sample collection from scratch whenever $L$, $\varepsilon$ or some transition costs change, DisCo can thus be *robust* to changes in such problem parameters. At the end of its exploration phase in Alg. 1, DisCo is able to perform zero-shot planning to solve other tasks restricted on $\mathcal{S}_L^{\rightarrow}$, such as cost-sensitive ones. Indeed in the following we show how the DisCo agent is able to compute an $\varepsilon/c_{\min}$-optimal policy for *any* stochastic shortest-path problem on $\mathcal{S}_L^{\rightarrow}$ with goal state $s \in \mathcal{S}_L^{\rightarrow}$ (i.e., $s$ is absorbing and zero-cost) and cost function lower bounded by $c_{\min} > 0$.

**Corollary 1.** *There exists an absolute constant $\beta > 0$ such that for any $L \geq 1$, $\varepsilon \in (0, 1]$ and $c_{\min} \in (0, 1]$ verifying $\varepsilon \leq \beta \cdot (L \, c_{\min})$, with probability at least $1 - \delta$, for* whatever *goal state $s \in \mathcal{S}_L^{\rightarrow}$ and* whatever *cost function $c$ in $[c_{\min}, 1]$, DisCo can compute (after its exploration phase, without additional environment interaction) a policy $\widehat{\pi}_{s,c}$ whose SSP value function $V_{\widehat{\pi}_{s,c}}$ verifies*

$$V_{\widehat{\pi}_{s,c}}(s_0 \to s) \leq V_{\mathcal{S}_L^{\rightarrow}}^{\star}(s_0 \to s) + \frac{\varepsilon}{c_{\min}},$$

*where $V_\pi(s_0 \to s) := \mathbb{E}\left[ \sum_{t=1}^{\tau_\pi(s_0 \to s)} c(s_t, \pi(s_t)) \,\big|\, s_1 = s_0 \right]$ is the SSP value function of a policy $\pi$ and $V_{\mathcal{S}_L^{\rightarrow}}^{\star}(s_0 \to s) := \min_{\pi \in \Pi(\mathcal{S}_L^{\rightarrow})} V_\pi(s_0 \to s)$ is the optimal SSP value function restricted on $\mathcal{S}_L^{\rightarrow}$.*

It is interesting to compare Cor. 1 with the reward-free exploration framework recently introduced by Jin et al. [24] in finite-horizon. At a high level, the result in Cor. 1 can be seen as a counterpart of [24] beyond finite-horizon problems, specifically in the goal-conditioned setting. While the parameter $L$ defines the horizon of interest for DisCo, resetting after every $L$ steps (as in finite-horizon) would prevent the agent to identify $L$-controllable states and lead to poor performance. This explains the distinct technical tools used: while [24] executes finite-horizon no-regret algorithms, DisCo deploys SSP policies restricted on the set of states that it "controls" so far. Algorithmically, both approaches seek to build accurate estimates of the transitions on a specific (unknown) state space of interest: the so-called "significant" states within $H$ steps for [24], and the incrementally $L$-controllable states $\mathcal{S}_L^{\rightarrow}$ for DisCo. Bound-wise, the cost-sensitive AX$^{\star}$ problem inherits the critical role of the minimum cost $c_{\min}$ in SSP problems (see App. C and e.g., [27, 28, 29]), which is reflected in the accuracy of Cor. 1 scaling inversely with $c_{\min}$. Another interesting element of comparison is the dependency on the size of the state space. While the algorithm introduced in [24] is robust w.r.t. states that can be reached with very low probability, it still displays a *polynomial* dependency on the total number of states $S$. On the other hand, DisCo has only a *logarithmic* dependency on $S$, while it directly depends on the number of $(L + \varepsilon)$-controllable states, which shows that DisCo effectively adapts to the state space of interest and it ignores all other states. This result is significant since not only $S_{L+\varepsilon}$ can be arbitrarily smaller than $S$, but also because the set $\mathcal{S}_{L+\varepsilon}^{\rightarrow}$ itself is initially unknown to the algorithm.

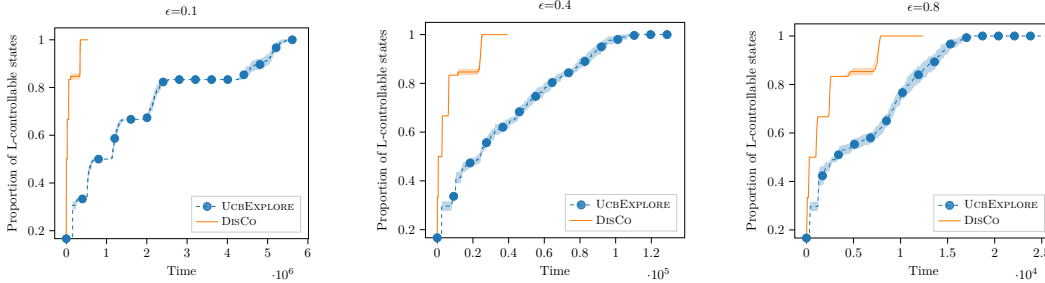

Figure 2: Proportion of the incrementally $L$-controllable states identified by `DisCo` and `UcbExplore` in a confusing chain domain for $L = 4.5$ and $\varepsilon \in \{0.1, 0.4, 0.8\}$. Values are averaged over 50 runs.

## 5 Numerical Simulation

In this section, we provide the first evaluation of algorithms in the incremental autonomous exploration setting. In the implementation of both `DisCo` and `UcbExplore`, we remove the logarithmic and constant terms for simplicity. We also boost the empirical performance of `UcbExplore` in various ways, for example by considering confidence intervals derived from the empirical Bernstein inequality (see [30]) as opposed to Hoeffding as done in [1]. We refer the reader to App. F for details on the algorithmic configurations and on the environments considered.

We compare the sample complexity empirically achieved by `DisCo` and `UcbExplore`. Fig. 2 depicts the time needed to identify all the incrementally $L$-controllable states when $L = 4.5$ for different values of $\varepsilon$, on a confusing chain domain. Note that the sample complexity is achieved soon after, when the algorithm can confidently discard all the remaining states as non-controllable (it is reported in Tab. 2 of App. F). We observe that `DisCo` outperforms `UcbExplore` for any value of $\varepsilon$. In particular, the gap in performance increases as $\varepsilon$ decreases, which matches the theoretical improvement in sample complexity from $\widetilde{O}(\varepsilon^{-3})$ for `UcbExplore` to $\widetilde{O}(\varepsilon^{-2})$ for `DisCo`. On a second environment — the combination lock problem introduced in [31] — we notice that `DisCo` again outperforms `UcbExplore`, as shown in App. F.

Another important feature of `DisCo` is that it targets the tighter objective AX$^\star$, whereas `UcbExplore` is only able to fulfill objective AX$_L$ and may therefore elect suboptimal policies. In App. F we show empirically that, as expected theoretically, this directly translates into higher-quality goal-reaching policies recovered by `DisCo`.

## 6 Conclusion and Extensions

**Connections to existing deep-RL methods.** While we primarily focus the analysis of `DisCo` in the tabular case, we believe that the formal definition of AX problems and the general structure of `DisCo` may also serve as a theoretical grounding of many recent approaches to unsupervised exploration. For instance, it is interesting to draw a parallel between `DisCo` and the ideas behind Go-Explore [32]. Go-Explore similarly exploits the following principles: (1) remember states that have previously been visited, (2) first return to a promising state (without exploration), (3) then explore from it. Go-Explore assumes that the world is deterministic and resettable, meaning that one can reset the state of the simulator to a previous visit to that cell. Very recently [15], the same authors proposed a way to relax this requirement by training goal-conditioned policies to reliably return to cells in the archive during the exploration phase. In this paper, we investigated the theoretical dimension of this direction, by provably learning such goal-conditioned policies for the set of incrementally controllable states.

**Future work.** Interesting directions for future investigation include: **1)** Deriving a lower bound for the AX problems; **2)** Integrating `DisCo` into the meta-algorithm MNM [33] which deals with incremental exploration for AX$_L$ in non-stationary environments; **3)** Extending the problem to continuous state space and function approximation; **4)** Relaxing the definition of incrementally controllable states and relaxing the performance definition towards allowing the agent to have a non-zero but limited sample complexity of learning a shortest-path policy for any state at test time.

## Broader Impact

This paper makes contributions to the fundamentals of online learning (RL) and due to its theoretical nature, we see no ethical or immediate societal consequence of our work.

## Footnotes

[1]We say that $f(\varepsilon) = \widetilde{O}(\varepsilon^\alpha)$ if there are constants $a$, $b$, such that $f(\varepsilon) \leq a \cdot \varepsilon^\alpha \log^b(\varepsilon)$.

[2]Lim and Auer [1] originally considered a countable, possibly infinite state space; however this leads to a technical issue in the analysis of UcbExplore (acknowledged by the authors via personal communication and explained in App. E.3), which disappears by considering only finite state spaces.

[3]This assumption should be contrasted with the finite-horizon setting, where each policy resets automatically after $H$ steps, or assumptions on the MDP dynamics such as ergodicity or bounded diameter, which guarantee that it is always possible to find a policy navigating between any two states.

[4]We refer the reader to [1, Sect. 2.1] for a more formal and complete characterization of this negative result.

[5]Note that we translated in the condition in [1] of a relative error of $L\varepsilon$ to an absolute error of $\varepsilon$, to align it with the common formulation of sample complexity in RL.

[6]In words, all actions at states in $\mathcal{K}_k$ behave exactly as in $M$ and suffer a unit cost, in all states outside $\mathcal{K}_k$ only the reset action to $s_0$ is available with a unit cost, and all actions at the goal $s'$ induce a zero-cost self-loop.

[7]Note that if we replace the error of $\varepsilon$ for $AX_L$ with an error of $L\varepsilon$ as in [1], we recover the sample complexity of $\widetilde{O}(L^3 S_{L+\varepsilon} A / \varepsilon^3)$ stated in [1, Thm. 8].

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
