[Supplementary Material]

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

1   Initialize $k := 0$, $K_0 := \{s_0\}$, $U_0 := \{\}$ and a restricted policy $\pi_{s_0} \in \Pi(K_0)$.
2   Set $\varepsilon := \min\{\varepsilon, 1\}$ and `continue := True`.
3   **while** `continue` **do**
4      Set $k \mathrel{+}= 1$. //new round
       // ¬ Sample collection on $K$
5      For each $(s,a) \in K_k \times A$, execute policy $\pi_s$ until the total number of visits $N_k(s,a)$ to $(s,a)$ satisfies $N_k(s,a) \geq n_k := \phi(K_k)$. For each $(s,a) \in K_k \times A$, add $s' \sim p(\cdot|s,a)$ to $U_k$ if $s' \notin K_k$.
       // ­ Restriction of candidate states $U$
6      Compute transitions $\widehat{p}_k(s'|s,a)$ and $W_k := \left\{ s' \in U_k : \exists (s,a) \in K_k \times A, \widehat{p}_k(s'|s,a) \geq \frac{1 - \varepsilon/2}{L} \right\}$
7      **if** $W_k$ is empty **then**
8          Set `continue := False`. //condition STOP1
9      **else**
         // ® Computation of the optimistic policies on $K$
10          **for** each state $s' \in W_k$ **do**
11            Compute $(\pi_{s'}, v_{s'}) := \text{OVI}_{\text{SSP}}(K_k, A, s', N_k, \frac{\varepsilon}{6L})$; see Alg. 3 in App. D.1.
12          Let $s^\dagger := \arg\min_{s \in W_k} v_s(s_0)$ and $v^\dagger := v_{s^\dagger}(s_0)$.
13          **if** $v^\dagger > L$ **then**

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

*with $\widehat{\Theta}(X) := \max_{(s,a) \in X \times A} \left( \sum_{s' \in X} \sqrt{\widehat{p}(s'|s,a)(1 - \widehat{p}(s'|s,a))} \right)^2$, then the algorithm DisCo (Alg. 1) satisfies the following sample complexity bound for AX$^\star$*

$$C_{\mathrm{AX}^\star}(\text{DisCo}, L, \varepsilon, \delta) = \widetilde{O}\left( \frac{L^5 \Gamma_{L+\varepsilon} S_{L+\varepsilon} A}{\varepsilon^2} + \frac{L^3 S_{L+\varepsilon}^2 A}{\varepsilon} \right), \tag{3}$$

*where $S_{L+\varepsilon} := |S_{L+\varepsilon}^{\rightarrow}|$ and*

$$\Gamma_{L+\varepsilon} := \max_{(s,a) \in S_{L+\varepsilon}' \times A} \left| \{ p(s'|s,a) \}_{s' \in S_{L+\varepsilon}'} \right|_0 \

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

# Appendix

## A  Autonomous Exploration Objectives

We recall the two $AX$ objectives stated in Def. 5: for any length $L \geq 1$, error threshold $\varepsilon > 0$ and confidence level $\delta \in (0,1)$, the sample complexities $C_{AX_L}(A; L; \varepsilon; \delta)$ and $C_{AX^?}(A; L; \varepsilon; \delta)$ are defined as the number of time steps required by a learning algorithm $A$ to identify a set $K \subseteq S_L^{\rightarrow}$ such that with probability at least $1 - \delta$, it has learned a set of policies $\{\pi_s\}_{s \in K}$ that respectively verifies the following $AX$ requirement

$$(AX_L) \quad \forall s \in K; v_{\pi_s}(s_0 \to s) \leq L + \varepsilon,$$
$$(AX^?) \quad \forall s \in K; v_{\pi_s}(s_0 \to s) \leq V^?_{S_L^{\rightarrow}}(s_0 \to s) + \varepsilon:$$

As we explain in Sect. 4, DisCo (Alg. 1) succeeds in tackling condition $AX^?$, whereas UcbExplore [1], which is designed to tackle condition $AX_L$, is unable to tackle $AX^?$. Note that the algorithmic design of UcbExplore entails that it computes policies whose value function implicitly targets $V^?_{K_t}$, with $K_t$ the current set of controllable states. While $V^?_{K_t}$ is always smaller than $L$, UcbExplore cannot provide any tightness guarantees w.r.t. $V^?_{K_t}$ since it has no guarantee that the transition dynamics are estimated well enough on $K_t$. An additional challenge with which UcbExplore fails to cope is the fact that the set $K_t$ increases over time and thus unlocks new states and paths, which may be useful to improve its shortest-path policies for previously discovered states.

To better understand this phenomenon, let us introduce an alternative condition $AX^0$ tighter than $AX_L$, but looser than $AX^?$ — which stems from the challenge of not knowing $S_L^{\rightarrow}$ in advance. We define $AX^0$ as follows: for any state $s$ in $S_L^{\rightarrow}$, the objective is to find a policy that can reach $s$ from $s_0$ in at most $L^0 + \varepsilon$ steps on average, where $L^0 := \min\{l \leq L : s \in S_l^{\rightarrow}\}$, i.e.,

$$(AX') \quad \forall s \in K; v_{\pi_s}(s_0 \to s) \leq L^0 + \varepsilon, \text{ where } L^0 := \min\{l \leq L : s \in S_l^{\rightarrow}\}.$$

As mentioned in [1, Corollary 9], it is possible to run separate instances of UcbExplore with increasing $L_n = 1 + n\varepsilon$ from $n = 0$ to $\lceil \frac{L-1}{\varepsilon} \rceil$ (i.e., until $n$ satisfies $L_{n-1} \leq L \leq L_n$). This verifies the condition $AX^0$ at the cost of a worsened dependency on both $\varepsilon$ and $L$ as follows

$$C_{AX^0}(\text{UcbExplore}; L; \varepsilon; \delta) = \tilde{O}\left(\frac{L^7 S_{L+\varepsilon} A}{\varepsilon^4}\right):$$

While $AX^0$ is tighter than $AX_L$, it may be arbitrarily loose compared to $AX^?$, which illustrates the intrinsic limitations in UcbExplore design. UcbExplore incrementally expands a set of "controllable" states $K$: starting with $K_0 = \{s_0\}$, at time $t$ a state $s$ is added to $K_t$ whenever UcbExplore can confidently assess that it managed to learn a policy reaching $s$ in less than $L$ steps. Since at time $t$ UcbExplore can only consider policies restricted to the controllable states $K_t$, even the shortest-path policy computed to reach $s$ at time $t$ may not be $\varepsilon$-optimal w.r.t. to the whole set $S_L^{\rightarrow}$. Indeed, every time a state is added to $K$, this state may unlock new paths which may, for previously controllable states, allow for better shortest-path policies restricted on the updated $K$. Fig. 3 illustrates this behavior, where the state $y$ unlocks a fast path from $s_0$ to $x$ which should be taken in $y$ instead of resetting to $s_0$. Consequently, if the agent seeks to tackle condition $AX^?$, it must have the faculty to backtrack, i.e., continuously update both its belief of the vicinity ($K$) and its notion of optimality on the vicinity ($V^?_K$). Unfortunately, UcbExplore can only compute policies targeting $V^?_K$ with $K$ the current set of controllable states, but it fails to be accurate enough to devise such policies as the set of

Figure 3: Let $X := \{s_0\} \cup \{x\}$ and $Y := X \cup \{y\}$. For any $l \geq 1$, suppose that from $s_0$, the agent reaches $x$ in $l$ steps with probability $1/2$, or reaches $y$ in $l + 1$ steps with probability $1/2$. If the goal state is $x$, constraining an agent to use policies restricted to $X$ (i.e., that reset to $s_0$ outside of $X$) is detrimental since $x$ can actually be reached in 1 step from $y$. Formally, we can easily prove that $V^?_X(s_0 \to x) - V^?_Y(s_0 \to x) = l + 1$, which grows arbitrarily as $l$ increases.

| AX | UcbExplore [1] | DisCo (Alg. 1) |
|---|---|---|
| $AX_L$ | $\tilde{O}\left(\dfrac{L^6 S_{L+\varepsilon} A}{\varepsilon^3}\right)$ | $\tilde{O}\left(\dfrac{L^5_{L+\varepsilon} S_{L+\varepsilon} A}{\varepsilon^2} + \dfrac{L^3 S^2_{L+\varepsilon} A}{\varepsilon}\right)$ |
| $AX^0$ | $\tilde{O}\left(\dfrac{L^7 S_{L+\varepsilon} A}{\varepsilon^4}\right)$ | |
| $AX^?$ | Unable | |

Table 1: Comparison between the sample complexity of UcbExplore and DisCo, depending on the condition $AX_L$, $AX^0$ or $AX^?$.

controllable state $x$ is expanded over time. In contrast, in virtue of its allocation function (Eq. 2) which enables to track the number of collected samples as $K$ increases, DisCo is able to improve its candidate shortest-path policies during the consolidation step when the final set $K$ is considered.

The following general and simple statement captures how the expansion of the state space of interest may alter and refine the optimal policy restricted on it.

Lemma 1. For any two sets $X \subseteq Y$ and any state $x \in X$, we have $V^?_X(s_0 \to x) \geq V^?_Y(s_0 \to x)$. Moreover, the gap between the two quantities may be arbitrarily large.

Proof. The inequality is immediate from Asm. 1. Fig. 3 shows the gap may be arbitrarily large. □

Finally, we summarize all the sample complexity results in Tab. 1.

# B  Efficient Computation of Optimistic SSP Policy

In this section we recall from [27, 28] how to efficiently compute an optimistic stochastic shortest-path (SSP) policy.

## B.1  Computation of Optimal Policy in Known SSP

This section details the procedure to efficiently compute an (arbitrarily near-) optimal policy in a known SSP instance with positive costs and which admits at least one proper policy. Recall that a proper policy is a policy whose execution starting from any non-goal state eventually reaches the goal state with probability one [26].

Definition 7 (SSP-MDP). An SSP-MDP is an MDP $M = (S^y; A; s^y; p; c)$ where $S^y$ is the set of non-goal states with $|S^y| = S^y$, $A$ is the set of actions, $p$ is the transition function and $c$ is the cost function. The goal state $s^y \notin S^y$ is zero-cost and absorbing, i.e., $p(s^y|s^y; a) = 1$ and $c(s^y; a) = 0$ for any $a \in A$.

The (possibly unbounded) value function (also called expected cost-to-go) of any policy $\pi$ starting from state $s_0$ is defined as

$$V^\pi(s_0) := E\left[\sum_{t=1}^{\infty} c(s_t; \pi(s_t)) \,\middle|\, s_0\right] = E\left[\sum_{t=1}^{\tau_\pi(s_0 \to s^y)} c(s_t; \pi(s_t)) \,\middle|\, s_0\right].$$

Assumption 2. We restrict the attention to SSP-MDP $M$ (see Def. 7) such that, for any $(s; a) \in S^y \times A$, $c(s; a) \in [c_{min}; 1]$ with $c_{min} > 0$. (Note that having positive costs ensures that for any non-proper policy $\pi$ there exists a state $s$ with $V^\pi(s) = +\infty$.) Moreover, we assume that there exists at least one proper policy (i.e., that reaches the goal state $s^y$ with probability one starting from any state in $S^y$).

The procedure $VI_{SSP}$ considers the following inputs: a goal $s^y$, non-goal states $S^y$, a known model $p$ and a known cost function $c$, with (non-goal) costs lower bounded by $c_{min} > 0$. $VI_{SSP}$ outputs a vector $u$ (of size $|S^y|$) and a policy $\pi$ which is greedy w.r.t. the vector $u$.

The optimal Bellman operator is defined as follows for any vector $u$ and any non-goal state $s \in S^y$

$$Lu(s) := \min_{a \in A}\left\{c(s; a) + \sum_{s^0 \in S^y} p(s^0|s; a) u(s^0)\right\}.$$

Algorithm 2: $\mathrm{VI}_{\mathrm{SSP}}$

---

Input: Non-goal states $S^{\neg g}$, action set $A$, transitions $p$, costs $c$ and accuracy $\epsilon$
Output: Value vector $u$ and greedy policy $\pi$

1   Define $Lu(s) := \min_{a \in A} \left\{ c(s,a) + \sum_{s' \in S^{\neg g}} p(s'|s,a) u(s') \right\}$
2   Set $u_0 = 0_{S^{\neg g}}$ and $j = 0$
3   $u_1 = L u_0$
4   while $\|u_{j+1} - u_j\|_1 > \epsilon$ do
5   |   $u_{j+1} = L u_j$
6   Set $u := u_j$ and $\pi(s) \in \arg\min_{a \in A} \left\{ c(s,a) + \sum_{s' \in S^{\neg g}} p(s'|s,a) u(s') \right\}$ for any $s \in S^{\neg g} [= S \setminus \{s^g\}]$

---

Note that by definition, $V_\pi(s^g) = 0$ for any $\pi$. We perform a value iteration ($VI$) scheme over this operator as explained in [e.g., 9, 34, 27]. Namely, we consider initial vector $u_0 := 0$ and set iteratively $u_{i+1} := L u_i$ (see Alg. 2). For a predefined precision $\epsilon > 0$, the stopping condition is reached for the first iteration $j$ such that $\|u_{j+1} - u_j\|_1 \leq \epsilon$. The policy is then selected to be the greedy policy w.r.t. the vector $u := u_j$, i.e.,

$$\forall s \in S^{\neg g} [= S \setminus \{s^g\}], \quad \pi(s) \in \arg\min_{a \in A} \left\{ c(s,a) + \sum_{s' \in S^{\neg g}} p(s'|s,a) u(s') \right\}. \tag{7}$$

Importantly, while $u$ is not the value function of $\pi$, both quantities can be related according to the following lemma.

Lemma 2. Consider an SSP-MDP $M = (S^{\neg g}; A; s^g; p; c)$ defined as in Def. 7 and satisfying Asm. 2. Let $(u; \pi) = \mathrm{VI}_{\mathrm{SSP}}(S^{\neg g}; A; p; c; \epsilon)$ be the solution computed by $\mathrm{VI}_{\mathrm{SSP}}$. Denote by $V_\pi$ the true value function of $\pi$ and by $V^\star = V_{\pi^\star} = L V^\star$ the optimal value function. The following component-wise inequalities hold

- $u \leq V^\star \leq V_\pi$.

- If the $VI$ precision level verifies $\epsilon \leq \frac{c_{\min}}{2}$, then $V_\pi \leq \left(1 + \frac{2\epsilon}{c_{\min}}\right) u$.

Proof. The result can be obtained by adapting [27, Lem. 4 & App. E]. For the first inequality, given that we consider the initial vector $u_0 = 0$, we know that $0 \leq V^\star$ with $V^\star = L V^\star$ by definition. By monotonicity of the operator $L$ [25, 26], we obtain $u_j \leq V^\star \leq V_\pi$. As for the second inequality, we introduce the following Bellman operators of a deterministic policy $\pi$ for any vector $u$ and state $s$,

$$L_\pi u(s) := c(s, \pi(s)) + \sum_{s' \in S} p(s'|s, \pi(s)) u(s');$$

$$T_\pi u(s) := \underbrace{c(s, \pi(s))}_{> 0} + \epsilon + \sum_{s' \in S} p(s'|s, \pi(s)) u(s').$$

Note that the SSP problem defined by the operator $T_\pi$ satisfies Asm. 2 since i) it has positive costs due to the condition $\epsilon \leq \frac{c_{\min}}{2}$ and ii) the fact that $M$ satisfies Asm. 2 guarantees the existence of at least one proper policy in the model. We can write component-wise

$$T_\pi u_j = L_\pi u_j \overset{(a)}{=} L u_j \overset{(b)}{\leq} u_j + \epsilon;$$

where (a) uses that $\pi$ is the greedy policy w.r.t. $u_j$ and (b) stems from the chosen stopping condition which yields $L u_j \leq u_j + \epsilon$. By monotonicity of the operator $T_\pi$, we have for all $m > 0$, $(T_\pi)^m u_j \leq u_j$. The asymptotic convergence of the operator in an SSP problem satisfying Asm. 2 (see e.g., [6, Prop. 2.2.1]) guarantees that taking the limit $m \to +1$ yields $W_\pi \leq u_j$, where $W_\pi$ is defined as the value function of policy $\pi$ in the model $p$ with $\epsilon$ subtracted to all the costs, i.e.,

$$W_\pi(s) := \mathbb{E}\left[ \sum_{t=1}^{\tau^{(s)}} (c(s_t, \pi(s_t)) - \epsilon) \,\middle|\, s_1 = s \right] = V_\pi(s) - \mathbb{E}[\tau(s)];$$

Figure 4: Optimistic Value Iteration for SSP ($\mathrm{OVI}_{\mathrm{SSP}}$).

where $\tau_\pi(s)$ denotes the (random) hitting time of policy $\pi$ to reach the goal starting from state $s$. Moreover, we have $c_{min} \mathbb{E}[\tau_\pi(s)] \leq V^\pi(s) \leq c_{max} \mathbb{E}[\tau_\pi(s)]$. Putting everything together, we thus get

$$\frac{1}{1 - \frac{\epsilon}{c_{min}}} V^\star \leq u_j. \text{ Since } \epsilon \leq \frac{c_{min}}{2}, \text{ we ultimately obtain}$$

$$V^\star \leq \frac{1}{1 - \frac{\epsilon}{c_{min}}} u_j \leq \left(1 + \frac{2\epsilon}{c_{min}}\right) u_j;$$

where the last inequality uses the fact that $\frac{1}{1-x} \leq 1 + 2x$ holds for any $0 \leq x \leq \frac{1}{2}$. □

### B.2 Computation of Optimistic Model in Unknown SSP

Consider an SSP problem $M$ defined as in Asm. 2. Consider that, at any given stage of the learning process, the agent is equipped with $N(s,a)$ samples at each state-action pair. A method to compute an optimistic model $\tilde{p}$ is provided in [28], which we recall below.

Denote by $\bar{p}$ the current empirical average of transitions $\bar{p}(s'|s,a) := N(s,a,s')/N(s,a)$, and set $\bar{\sigma}^2(s'|s,a) := \bar{p}(s'|s,a)(1 - \bar{p}(s'|s,a))$ as well as $N^+(s,a) := \max\{1, N(s,a)\}$. For any $(s,a,s') \in S^y \times A \times S^y$, the empirical Bernstein inequality [35, 36] is leveraged to select the following confidence intervals (with probability at least $1-\delta$) on the transition probabilities

$$\beta(s,a,s') := 2\sqrt{\frac{\bar{\sigma}^2(s'|s,a)}{N^+(s,a)} \log\left(\frac{2SAN^+(s,a)}{\delta}\right)} + \frac{6 \log\left(\frac{2SAN^+(s,a)}{\delta}\right)}{N^+(s,a)};$$

and $\beta(s,a,s^y) := \sum_{s' \in S^y} \beta(s,a,s')$. The selection of the optimistic model $\tilde{p}$ is as follows: the probability of reaching the goal $s^y$ is maximized at every state-action pair, which implies minimizing the probability of reaching all other states and setting them at the lowest value of their confidence range. Formally, we set for all $(s,a,s') \in S^y \times A \times S^y$,

$$\tilde{p}(s'|s,a) := \max\left\{\bar{p}(s'|s,a) - \beta(s,a,s'); 0\right\};$$

and $\tilde{p}(s^y|s,a) := 1 - \sum_{s' \in S^y} \tilde{p}(s'|s,a)$.

### B.3 Combining the two: Optimistic Value Iteration for SSP ($\mathrm{OVI}_{\mathrm{SSP}}$)

$\mathrm{OVI}_{\mathrm{SSP}}$ first computes an optimistic model $\tilde{p}$ leveraging App. B.2, and it then runs the $\mathrm{VI}_{\mathrm{SSP}}$ procedure of App. B.1 in the model $\tilde{p}$, i.e., $(\tilde{u}, e) = \mathrm{VI}_{\mathrm{SSP}}(S^y; A; s^y; \tilde{p}; c)$. This outputs an optimistic pair $(\tilde{u}, e)$ composed of the VI vector $\tilde{u}$ and the policy $e$ that is greedy w.r.t. $\tilde{u}$ in the model $\tilde{p}$. The $\mathrm{OVI}_{\mathrm{SSP}}$ scheme is recapped in Fig. 4.

## C   Useful Result: Simulation Lemma for SSP

Consider a stochastic shortest-path (SSP) instance (see Def. 7) that satisfies Asm. 2. We denote by $A = |A|$ the number of actions, $S = |S|$ the number of non-goal states, $g \notin S$ the (zero-cost and absorbing) goal state, $p$ the unknown transitions and $c$ the known cost function. We assume that $0 < c(s,a) \leq 1$ for all $(s,a) \in S \times A$, and set $c_{min} := \min_{s,a} c(s,a) > 0$. We also set $S^0 := S \cup \{g\}$.

Recall that the goal state is zero-cost (i.e., $c(g; a) = 0$) and absorbing (i.e., $p(g|g; a) = 1$), and that the value function of a policy amounts to the expected cumulative costs following this policy until reaching the goal.

Definition 8. For any model $p$ and $\epsilon > 0$, we introduce the set of models close to $p$ w.r.t. the $\ell_1$-norm on the non-goal states as follows

$$\mathcal{P}^{(p)} := \left\{ p^0 \in \mathbb{R}^{S^0 \times A \times S^0} : \ \forall (s; a) \in S \times A \ ; \ p^0(\cdot | s; a) \in \Delta(S^0); \ p(g|g; a) = 1 \ ; \right.$$
$$\left. \sum_{y \in S} |p(y|s; a) - p^0(y|s; a)| \leq \epsilon \right\}.$$

Lemma 3 (Simulation Lemma for SSP). Consider any model $p$ and $p^0 \in \mathcal{P}^{(p)}$ such that, for each model, there exists at least one proper policy w.r.t. the goal state $g$. Consider any policy $\pi$ that is proper in $p^0$, with value function denoted by $V^0$, such that the following condition is verified

$$\epsilon \| V^0 \|_1 \leq 2 c_{min} : \tag{8}$$

Then $\pi$ is proper in $p$ (i.e., its value function verifies $V < +\infty$ component-wise), and we have

$$\forall s \neq g; \quad V(s) \leq \left(1 + \frac{2 \epsilon \| V^0 \|_1}{c_{min}}\right) V^0(s);$$

and conversely,

$$\forall s \neq g; \quad V^0(s) \leq \left(1 + \frac{\epsilon \| V^0 \|_1}{c_{min}}\right) V(s):$$

Combining the two inequalities above yields

$$\| V - V^0 \|_1 \leq \frac{7 \epsilon \| V^0 \|_1^2}{c_{min}}:$$

Proof. The proof of Lem. 3 requires a result of [7] recalled in Lem. 4 and can be seen as a generalization of [28, Lem. B.4]. First, let us assume that $\pi$ is proper in the model $p^0$. This implies that its value function, denoted by $V^0$, is bounded component-wise. Moreover, for any non-goal state $s \in S$, the Bellman equation holds as follows

$$V^0(s) = c(s; \pi(s)) + \sum_{y \in S} p^0(y|s; \pi(s)) V^0(y)$$
$$= c(s; \pi(s)) + \sum_{y \in S} p(y|s; \pi(s)) V^0(y) + \sum_{y \in S} (p^0(y|s; \pi(s)) - p(y|s; \pi(s))) V^0(y): \tag{9}$$

By successively using Hölder's inequality and the facts that $p^0 \in \mathcal{P}^{(p)}$ and $c(s; \pi(s)) \geq c_{min}$, we get

$$V^0(s) \leq c(s; \pi(s)) + \epsilon \| V^0 \|_1 + p(\cdot | s; \pi(s))^\top V^0 \leq c(s; \pi(s)) \left(1 + \frac{\epsilon \| V^0 \|_1}{c_{min}}\right) + p(\cdot | s; \pi(s))^\top V^0:$$

Let us now introduce the vector $V^{00} := \left(1 - \frac{\epsilon \| V^0 \|_1}{c_{min}}\right)^{-1} V^0$. Then for all $s \in S$,

$$V^{00}(s) \geq c(s; \pi(s)) + p(\cdot | s; \pi(s))^\top V^{00}:$$

Hence, from Lem. 4, $\pi$ is proper in $p$ (i.e., $V < +\infty$), and we have

$$V \leq V^{00} \leq \left(1 + 2 \frac{\epsilon \| V^0 \|_1}{c_{min}}\right) V^0; \tag{10}$$

where the last inequality stems from condition (8) and the fact that $\frac{1}{1-x} \leq 1 + 2x$ holds for any $0 \leq x \leq \frac{1}{2}$. Conversely, analyzing Eq. 9 from the other side, we get

$$V^0(s) \geq c(s; \pi(s)) \left(1 + \frac{\epsilon \| V^0 \|_1}{c_{min}}\right) + p(\cdot | s; \pi(s))^\top V^0:$$

Let us now introduce the vector $V^{00} := \left(1 + \frac{kV^0 k_1}{c_{min}}\right)^{-1} V^0$. Then

$$V^{00}(s) \geq c(s, \pi(s)) + p(\cdot | s, \pi(s))^\top V^{00}.$$

We then obtain in the same vein as Lem. 4 (by leveraging the monotonicity of the Bellman operator $L_\pi U(s) := c(s, \pi(s)) + p(\cdot | s, \pi(s))^\top U$) that $V^{00} \geq V_\pi$, and therefore

$$V^0 \geq \left(1 + \frac{kV^0 k_1}{c_{min}}\right) V_\pi. \tag{11}$$

Combining Eq. 10 and 11 yields component-wise

$$kV_\pi - V^0 k_1 \leq 2\frac{kV^0 k_1}{c_{min}} kV^0 k_1 + \frac{kV^0 k_1}{c_{min}} kV_\pi k_1 \leq 7\frac{kV^0 k_1^2}{c_{min}};$$

where the last inequality uses that $kV_\pi k_1 \leq 5kV^0 k_1$ which stems from plugging condition (6) into Eq. 10.

Note that here $p$ and $p^0$ play symmetric roles; we can perform the same reasoning in the case where $\pi$ is proper in the model $p$ and it would yield an equivalent result by switching the dependencies on $V$ and $V^0$. $\square$

**Lemma 4** ([37], Lem. 1). *In an SSP-MDP satisfying Asm. 2, let $\pi$ be any policy, then*
- *If there exists a vector $U : S \to \mathbb{R}$ such that $U(s) \geq c(s, \pi(s)) + \sum_{s' \in S} p(s'|s, \pi(s)) U(s')$ for all $s \in S$, then $\pi$ is proper, and $V_\pi$ the value function of $\pi$ is upper bounded by $U$ component-wise, i.e., $V_\pi(s) \leq U(s)$ for all $s \in S$.*
- *If $\pi$ is proper, then its value function $V_\pi$ is the unique solution to the Bellman equations $V_\pi(s) = c(s, \pi(s)) + \sum_{s' \in S} p(s'|s, \pi(s)) V_\pi(s')$ for all $s \in S$.*

## D  Proof of Theorem 1 (Sample Complexity Analysis of DisCo)

### D.1  Computation of the Optimistic Policies

At each round $k$, for each goal state $y \in W_k$, DisCo computes an optimistic goal-oriented policy associated to the MDP $M_k^0(s^y)$ constructed as in Def. 6. This MDP is defined over the entire state space $S$ and restricts the action to the only action RESET outside $K_k$. We can build an equivalent MDP by restricting the focus on $K_k$. To this end, we define the following SSP-MDP.

**Definition 9.** *Define $M_k^y(s^y) := \langle S_k^y; A_k^y(\cdot); c_k^y; p_k^y \rangle$ where $S_k^y := K_k \cup \{f s^y; x g\}$ and $S_k^y = |S_k^y| = |K_k| + 2$. State $x$ is a meta-state that encapsulates all the states that have been observed so far and are not in $K_k$. The action space $A_k^y(\cdot)$ is such that $A_k^y(s) = A$ for all states $s \in K_k$ and $A_k^y(s) = \{RESET\}$ for $s \in \{s^y; x\}$. The cost function is $c_k^y(x; a) = 0$ for any $a \in A_k^y(x)$ and $c_k^y(s; a) = 1$ everywhere else. The transition function is defined as $p_k^y(s^y | s^y; a) = p_k^y(s_0 | x; a) = 1$ for any $a$, $p_k^y(y | s; a) = p(y | s; a)$ for any $(s; a; y) \in K_k \times A \times (K_k \cup \{s^y g\})$ and $p_k^y(x | s; a) = 1 - \sum_{y \in K_k \cup \{f s^y g\}} p_k^y(y | s; a)$.*

Note that solving $M_k^y$ yields a policy effectively restricted to the set $K_k$ insofar as we can interpret the meta-state $x$ as $S \setminus \{K_k \cup \{f s^y gg$. Since $p$ is unknown, we cannot construct $M_k^y(s^y)$. Let $N_k$ be the state-action counts accumulated up until now. We denote by $\hat{p}_k$ the "global" empirical estimates, i.e., $\hat{p}_k(y | s; a) = N_k(s; a; y)/N_k(s; a)$. Given them, we define the "restricted" empirical estimates $\hat{p}_k^y$ as follows: $\hat{p}_k^y(y | s; a) := \hat{p}_k(y | s; a)$ for any $(s; a; y) \in K_k \times A \times (K_k \cup \{f s^y g\})$ and $\hat{p}_k^y(x | s; a) := 1 - \sum_{y \in K_k \cup \{f s^y g\}} \hat{p}_k^y(y | s; a)$. Denoting $N_k^+(s; a) := \max\{1; N_k(s; a)g\}$, we then define the following bonuses for any $(s; a; y) \in K_k \times A \times (K_k \cup \{f s^y g\})$,

$$\beta_k(s; a; y) := 2\sqrt{\frac{\hat{p}_k(y | s; a)(1 - \hat{p}_k(y | s; a))}{N_k^+(s; a)} \log \frac{2SAN_k^+(s; a)}{\delta}} + \frac{6 \log \frac{2SAN_k^+(s;a)}{\delta}}{N_k^+(s; a)}; \tag{12}$$

$$\gamma_k(s; a; x) := \sum_{y \in K_k \cup \{f s^y g\}} \beta_k(s; a; y): \tag{13}$$

---
**Algorithm 3:** $\text{OVI}_{\text{SSP}}$

---
**Input:** $K_k, A, s^y, N_k, \epsilon > 0$
**Output:** Value vector $v^y$ and policy $e^y$

1   Estimate transitions probabilities $\bar{p}_k$ using $N_k$
2   Compute the optimistic SSP-MDP $\tilde{M}_k^y$ as detailed in Def. 10
3   Compute $(v_k^y; e_k^y) = \text{VI}_{\text{SSP}}(S_k^y; A_k^y; c_k^y; \tilde{p}_k^y; \epsilon)$ (see Alg. 2)

---

Moreover, we set the uncertainty about the MDP at the meta-state $x$ and at the goal state $s^y$ to 0 by construction (since their outgoing transitions are deterministic, respectively to $s_0$ and $s^y$).

We now leverage the optimistic construction mentioned in App. B.1.

**Definition 10.** We denote by $\tilde{M}_k^y(s^y) = \langle S_k^y; A_k^y(\cdot); c_k^y; \tilde{p}_k^y \rangle$ the optimistic MDP associated to $M_k^y(s^y)$ defined in Def. 9. Then $\forall (s; a) \in K_k \times A$,

$$\tilde{p}_k^y(y|s; a) := \max \{ \bar{p}_k(y|s; a) - \beta_k(s; a; y); 0 \}; \quad \forall y \in K_k [ f \ x g; \tag{14}$$

$$\tilde{p}_k^y(s^y|s; a) := 1 - \sum_{y \in K_k [ f \ x g} \tilde{p}_k^y(y|s; a); \tag{15}$$

$$\tilde{p}_k^y(s^y|s^y; a) = \tilde{p}_k^y(s_0|x; a) = 1 : \tag{16}$$

Given this MDP, we can compute the optimistic value vector $v_k^y$ and policy $e_k^y$ using value iteration for SSP: $(v_k^y; e_k^y) = \text{VI}_{\text{SSP}}(S_k^y; A_k^y; c_k^y; \tilde{p}_k^y; \frac{\epsilon}{4L})$. We summarize the construction of the optimistic model and the computation of value function and policy in Alg. 3 ($\text{OVI}_{\text{SSP}}$).

**Remark.** Note that the structure of the problem does not appear to allow for variance-aware improvements in the analysis of Thm. 1 (specifically, when the analysis will apply an SSP simulation lemma argument). Indeed, given the possibly large number of states in the total environment $S$, the computation of the optimistic policies requires the construction of the meta-state $x$ that encapsulates all the states in $S \setminus f K_k [ f \ s^y gg$, where $s^y$ is the candidate goal state considered at round $k$. As a result, the uncertainty on the transitions reaching $x$ needs to be summed over multiple states, as shown in Eq. 13. This extra uncertainty at a single state in the induced MDP has the effect of canceling out Bernstein techniques seeking to lower the prescribed requirement of the state-action samples that the algorithm should collect. In turn this implies that such variance-aware techniques would not lead to any improvement in the final sample complexity bound.

### D.2  High-Probability Event

**Lemma 5.** It holds with probability at least $1 - \delta$ that for any time step $t \geq 1$ and for any state-action pair $(s; a)$ and next state $s'$,

$$j \bar{p}_t(s'|s; a) - p(s'|s; a)j \leq 2 \sqrt{ \frac{\tilde{b}_t^2(s'|s; a)}{N_t^+(s; a)} \log \frac{2SAN_t^+(s; a)}{\delta} } + \frac{6 \log \frac{2SAN_t^+(s;a)}{\delta}}{N_t^+(s; a)}; \tag{17}$$

where $N_t^+(s; a) := \max \{ 1; N_t(s; a) \}$ and where $\tilde{b}_t^2$ are the population variance of transitions, i.e., $\tilde{b}_t^2(s'|s; a) := \bar{p}_t(s'|s; a)(1 - \bar{p}_t(s'|s; a))$.

**Proof.** The confidence intervals in Eq. 17 are constructed using the empirical Bernstein inequality, which guarantees that the considered event holds with probability at least $1 - \delta$, see e.g., [38]. $\square$

Define the set of plausible transition probabilities as

$$C_k^y := \bigcap_{(s;a) \in S_k^y \times A} C_k^y(s; a);$$

where
$$C_k^y(s,a) := \{ p \in C \mid p(\cdot \mid s^y, a) = 1_{s^y}, p(\cdot \mid x, a) = 1_{s_0}, |p(s^0 \mid s, a) - \bar{p}_k(s^0 \mid s, a)| \le \beta_k(s,a,s^0)\};$$

with $C$ the $S_k^y$-dimensional simplex and $\bar{p}_k$ the empirical average of transitions.

**Lemma 6.** Introduce the event $\mathcal{E} := \bigcap_{k=1}^{T+1}\bigcap_{s^y \in W_k}^{T} \{ p_k^y \in C_k^y \}$. Then $P(\mathcal{E}) \ge 1 - \frac{\delta}{3}$.

**Proof.** We have with probability at least $1 - \frac{\delta}{3}$ that, for any $y \ne x$, $|p_k^y(y \mid s, a) - \bar{p}_k^y(y \mid s, a)| \le \beta_k(s,a,y)$ from the empirical Bernstein inequality (see Eq. 17), and moreover

$$|\bar{p}_k^y(x \mid s, a) - p_k^y(x \mid s, a)| = |1 - \sum_{y \in 2K_k \setminus \{s^y\}} \bar{p}_k^y(y \mid s, a) - 1 - \sum_{y \in 2K_k \setminus \{s^y\}} p_k^y(y \mid s, a)|$$
$$\le \sum_{y \in 2K_k \setminus \{s^y\}} |p_k^y(y \mid s, a) - \bar{p}_k^y(y \mid s, a)| \le \beta_k(s,a,x). \qquad \square$$

**Lemma 7.** Under the event $\mathcal{E}$, for any round $k$ and any goal state $s^y \in W_k$, the optimistic model $\tilde{p}_k^y$ constructed in Def. 10 verifies $\tilde{p}_k^y \in P_k^{(p_k^y)}$, with $\gamma_k := 4\beta_k(s,a,x)$ where $\beta_k$ is defined in Eq. 13.

**Proof.** Combining the construction in Def. 10, the proof of Lem. 6 and the triangle inequality yields

$$\sum_{y \in 2K_k \setminus \{x\}} |\tilde{p}_k^y(y \mid s, a) - p_k^y(y \mid s, a)| \le \sum_{y \in 2K_k \setminus \{x\}} |\tilde{p}_k^y(y \mid s, a) - \bar{p}_k^y(y \mid s, a)| + |\bar{p}_k^y(y \mid s, a) - p_k^y(y \mid s, a)|$$
$$\le \sum_{y \in 2K_k \setminus \{x\}} \beta_k(s,a,y) + 2\beta_k(s,a,x)$$
$$\le 4\beta_k(s,a,x). \qquad \square$$

Throughout the remainder of the proof, we assume that the event $\mathcal{E}$ holds.

## D.3 Properties of the Optimistic Policies and Value Vectors

We recall notation. Let us fix any round $k$ and any goal state $s^y \in W_k$. We denote by $\tilde{\pi}_k^y$ the greedy policy w.r.t. $\tilde{v}_k^y(\cdot \to s^y)$ in the optimistic model $\tilde{p}_k^y$. Let $\tilde{v}_k^y(s \to s^y)$ be the value function of policy $\tilde{\pi}_k^y$ starting from state $s$ in the model $\tilde{p}_k^y$. We can apply Lem. 2 given that the conditions of Asm. 2 hold (indeed, we have $c_{min} = 1 > 0$ and there exists at least one proper policy to reach the goal state $s^y$ since it belongs to $W_k$). Moreover, we have that $\tilde{V}_{K_k}^\star(s_0 \to s^y) \le V_{K_k}^\star(s_0 \to s^y)$ given the way the optimistic model $\tilde{p}_k^y$ is computed (i.e., by maximizing the probability of transitioning to the goal at any state-action pair), see [28, Lem. B.12]. Hence we get the two following important properties.

**Lemma 8.** For any round $k$, goal state $s^y \in W_k$ and state $s \in K_k \setminus \{x\}$, we have under the event $\mathcal{E}$,

$$\tilde{v}_k^y(s \to s^y) \le V_{K_k}^\star(s \to s^y).$$

**Lemma 9.** For any round $k$, goal state $s^y \in W_k$ and state $s \in K_k \setminus \{x\}$, we have

$$v_k^y(s \to s^y) \le (1 + 2\gamma)\tilde{v}_k^y(s \to s^y).$$

## D.4 State Transfer from U to K (step ⁻)

We fix any round $k$ and any goal state $s^y \in W_k$ that is added to the set of "controllable" states $K$, i.e., for which $\tilde{v}_k^y(s_0 \to s^y) \le L$.

**Lemma 10.** Under the event $\mathcal{E}$, we have both following inequalities

$$\begin{cases} v_k^y(s_0 \to s^y) \le L + \varepsilon; \\ v_k^y(s_0 \to s^y) \le V_{K_k}^\star(s_0 \to s^y) + \varepsilon. \end{cases}$$

In particular, the first inequality entails that $s^y \in S_{L+\varepsilon}^\to$, which justifies the validity of the state transfer from $U$ to $K$.

Proof. We have

$$V_k^\pi(s_0 \to s^\dagger) \overset{(a)}{\leq} (1 + 2\epsilon)\hat{V}_k^\pi(s_0 \to s^\dagger) \overset{(b)}{<} \begin{cases} L + \frac{\varepsilon}{3} \\ \overset{(c)}{:} \ V_{K_k}^\star(s_0 \to s^\dagger) + \frac{\varepsilon}{3}; \end{cases} \tag{18}$$

where inequality (a) comes from Lem. 9, inequality (b) combines the algorithmic condition $\hat{V}_k^\pi(s_0 \to s^\dagger) \leq L$ and the VI precision level $\epsilon := \frac{\varepsilon}{6L}$, and finally inequality (c) combines Lem. 8 and the VI precision level. Moreover, for any state $s \in K_k$,

$$V_k^\pi(s \to s^\dagger) \overset{(a)}{\leq} V_{K_k}^\star(s \to s^\dagger) + \frac{\varepsilon}{3} \overset{(b)}{\leq} V_{K_k}^\star(s_0 \to s^\dagger) + 1 + \frac{\varepsilon}{3} \leq V_k^\pi(s_0 \to s^\dagger) + 1 + \frac{\varepsilon}{3};$$

where (a) comes from Lem. 8 and (b) stems from the presence of the RESET action (Asm. 1).

We now provide the exact choice of allocation function $\iota$ in Alg. 1. We introduce

$$\iota := \frac{2\varepsilon}{12(L + 1 + \varepsilon)(L + \frac{\varepsilon}{3})}:$$

(Note that $\iota = O(\varepsilon/L^2)$.) We set the following requirement of samples for each state-action pair $(s, a)$ at round $k$,

$$n_k = \phi(K_k) = \left\lceil \frac{57X_k^2}{\iota^2} \log\left(\frac{8eX_k\sqrt{2SA}}{\iota\sqrt{\delta}}\right)^2 + \frac{24|S_k^\dagger|}{\iota} \log\left(\frac{24|S_k^\dagger|SA}{\delta}\right)^3 \right\rceil; \tag{19}$$

where we define

$$X_k := \max_{(s,a) \in S_k^\dagger \times A} \sum_{s' \in S_k^\dagger} \sqrt{b_k^2(s'|s,a)};$$

with $b_k^2(s'|s,a) := \hat{p}_k^\pi(s'|s,a)(1 - \hat{p}_k^\pi(s'|s,a))$ the estimated variance of the transition from $(s,a)$ to $s'$. Leveraging the empirical Bernstein inequality (Lem. 5) and performing simple algebraic manipulations (see e.g. [39, Lem. 8 and 9]) yields that $\iota_k(s;a;x) \geq \iota$. From Lem. 7, this implies that $\hat{p}_k^\pi \in P^{(p_k^\pi)}$ with $\beta := 4\iota$. We can then apply Lem. 3 (whose condition 8 is verified), which gives

$$V_k^\pi(s_0 \to s^\dagger) \leq 1 + \|V_k^\pi(\cdot \to s^\dagger)\|_\infty \, \beta \cdot V_k^\pi(s_0 \to s^\dagger) \tag{20}$$

$$\leq (1 + \beta(L + 1 + \varepsilon))V_k^\pi(s_0 \to s^\dagger)$$

$$\leq V_k^\pi(s_0 \to s^\dagger) + \frac{2\varepsilon}{3};$$

where the last inequality uses that $\beta(L + 1 + \varepsilon)(L + \frac{\varepsilon}{3}) = \frac{2\varepsilon}{3}$ by definition of $\iota$. Plugging in Eq. 18 yields the sought-after inequalities.

□

## D.5 Termination of the Algorithm

**Lemma 11** (Variant of Lem. 17 of [1]). Suppose that for every state $s \in S$, each action $a \in A$ is executed $\phi \geq d L \log\left(\frac{3ALS}{\delta}\right)$ e times. Let $S_{s;a}^0$ be the set of all next states visited during the $\phi$ executions of $(s,a)$. Denote by $\Omega$ the complementary of the event

$$\bar{\Omega}\{(s', s, a) \in S^2 \times A : p(s'|s,a) \geq \frac{1}{L} \wedge s' \notin S_{s;a}^0\}:$$

Then $P(\Omega) \geq 1 - \frac{\delta}{3}$.

**Lemma 12.** Under the event $\Omega \setminus \Theta$, for any round $k$, either $S_L^\dagger \subseteq K_k$, or there exists a state $s^\dagger \in S_L^\dagger \cap K_k$ such that $s^\dagger \in W_k$ and is $L$-controllable with a policy restricted to $K_k$. Moreover, $|W_k| \leq 2LA|K_k|$.

**Proof of Lem. 12.** Consider a round $k$ such that $\mathcal{S}_L^! \cap \mathcal{K}_k$ is non-empty. Due to the incremental construction of the set $\mathcal{S}_L^!$ (Def. 4), there exists a state $s^y \in \mathcal{S}_L^!$ and a policy restricted to $\mathcal{K}_k$ that can reach $s^y$ in at most $L$ steps (in expectation). Hence there exists a state-action pair $(s, a) \in \mathcal{K}_k \times \mathcal{A}$ such that $p(s^y | s, a) \geq \frac{1}{L}$. Since $n(\mathcal{K}_k) \geq \lceil L \log \frac{3ALS}{\delta} \rceil$ samples are available at each state-action pair, according to Lem. 11, we get that, under the event $\mathcal{E}$, $s^y$ is found during the sample collection procedure for the state-action pair $(s, a)$ (step $\neg$), which implies that $s^y \in U_k$.

Moreover, the choice of allocation function $n$ guarantees in particular that there are more than $\left( \frac{4L^2}{\varepsilon^2} \log(\frac{2LSA}{\delta}) \right)$ samples available at each state-action pair $(s, a) \in \mathcal{K}_k \times \mathcal{A}$. From the empirical Bernstein inequality of Eq. 17, we thus have that $|p(s^y | s, a) - \hat{p}_k(s^y | s, a)| \leq \frac{\varepsilon}{2L}$ under the event $\mathcal{E}$. Consequently we have

$$\hat{p}_k(s^y | s, a) \geq \frac{1}{L} - |p(s^y | s, a) - \hat{p}_k(s^y | s, a)| \geq \frac{1 - \frac{\varepsilon}{2}}{L};$$

which implies that $s^y \in W_k$. Furthermore, we can decompose $W_k$ the following way

$$W_k = \bigcup_{(s, a) \in \mathcal{K}_k \times \mathcal{A}} Y_k(s, a);$$

where we introduce the subset

$$Y_k(s, a) := \left\{ s^0 \in U_k : \hat{p}_k(s^0 | s, a) \geq \frac{1 - \frac{\varepsilon}{2}}{L} \right\}.$$

We then have

$$1 = \sum_{s^0 \in S} \hat{p}_k(s^0 | s, a) \geq \sum_{s^0 \in Y_k(s, a)} \hat{p}_k(s^0 | s, a) \geq \frac{1 - \frac{\varepsilon}{2}}{L} |Y_k(s, a)|.$$

We conclude the proof by writing that

$$|W_k| \leq \sum_{(s, a) \in \mathcal{K}_k \times \mathcal{A}} |Y_k(s, a)| \leq \frac{L}{1 - \frac{\varepsilon}{2}} A |\mathcal{K}_k| \leq 2LA |\mathcal{K}_k|;$$

where the last inequality uses that $\varepsilon \leq 1$ (from line 2 of Alg. 1). $\qquad\square$

**Lemma 13.** Under the event $\mathcal{E} \cap \mathcal{F}$, when either condition **STOP1** or **STOP2** is triggered (at a round indexed by $K$), we have $\mathcal{S}_L^! \subseteq \mathcal{K}_K$.

**Proof.** If condition **STOP1** is triggered, Lem. 12 immediately guarantees that $\mathcal{S}_L^! \subseteq \mathcal{K}_K$ under the event $\mathcal{E}$. If condition **STOP2** is triggered, we have for all $s \in W_K$, $\hat{v}_s(s_0 \to s) > L$. From Lem. 8 this means that, under the event $\mathcal{F}$, for all $s \in W_K$, $V_{\pi_K}^?(s_0 \to s) > L$. Hence none of the states in $W_K$ can be reached in at most $L$ steps (in expectation) with a policy restricted to $\mathcal{K}_Q$. We conclude the proof using Lem. 12. $\qquad\square$

**Lemma 14.** Under the event $\mathcal{E} \cap \mathcal{F}$, when **DisCo** terminates at round $K$, for any state $s \in \mathcal{K}_K$, the policy $\pi_s$ computed during step $\flat$ verifies

$$v_{\pi_s}(s_0 \to s) \leq \min_{\pi \in \Pi(\mathcal{S}_L^!)} v_\pi(s_0 \to s) + \varepsilon.$$

Moreover, we have that $\mathcal{S}_L^! \subseteq \mathcal{K}_K \subseteq \mathcal{S}_{L+\varepsilon}^!$.

**Proof.** Assume that the event $\mathcal{E} \cap \mathcal{F}$ holds. Then when the final set $\mathcal{K}_K$ is considered and the new policies are computed using all the samples, Lem. 10 yields for all $s \in \mathcal{K}_K$,

$$v_{\pi_s}(s_0 \to s) \leq \min_{\pi \in \Pi(\mathcal{K}_K)} v_\pi(s_0 \to s) + \varepsilon.$$

Moreover Lem. 13 entails that $\mathcal{K}_K \supseteq \mathcal{S}_L^!$. This implies from Lem. 1 that

$$\min_{\pi \in \Pi(\mathcal{K}_K)} v_\pi(s_0 \to s) \leq \min_{\pi \in \Pi(\mathcal{S}_L^!)} v_\pi(s_0 \to s);$$

which means that $\mathcal{K}_K \subseteq \mathcal{S}_{L+\varepsilon}^!$. $\qquad\square$

## D.6 High Probability Bound on the Sample Collection Phase (step ?)

Denote by $K$ the (random) index of the last round during which the algorithm terminates. We focus on the sample collection procedure for any state $s \in \mathcal{K}_K$. We denote by $k_s$ the index of the round during which $s$ was added to the set of "controllable" states $\mathcal{K}$. To collect samples at state $s$, the learner uses the shortest-path policy $\pi_s$. We say that an attempt to collect a specific sample is a rollout. We denote by $Z_K := |\mathcal{K}_K| A N_K$ the total number of samples that the learner needs to collect. As such, at most $Z_K$ rollouts must take place. Assume that the event holds. Then from Lem. 14, we have $\mathcal{K}_K \subseteq S^?_{L+\epsilon}$. Hence, denoting $S_{L+\epsilon} := |S^?_{L+\epsilon}|$, we have $Z_K \le Z_{L+\epsilon} := S_{L+\epsilon} A (S^?_{L+\epsilon})$. The following lemma provides a high-probability upper bound on the time steps required to meet the sampling requirements.

**Lemma 15.** Assume that the event holds. Set

$$\Pi := 4(L+\epsilon+1)\log\frac{6Z_{L+\epsilon}}{\delta};$$

and introduce the following event

$$T := \Big\{ \exists \text{ one rollout (with goal state } s) \text{ s.t. } \tau_s(s_0 \to s) > \Pi \Big\}:$$

We have $P(T) \le \frac{\delta}{3}$.

*Proof.* Assume that the event holds. Leveraging a union bound argument and applying Lem. 16 to policy $\pi_s$ which verifies $v_{\pi_s}(s^0 \to s) \le L+\epsilon+1$ for any $s^0 \in \mathcal{K}_{k_s}$, we get

$$P(T) \le \sum_{\text{rollouts}} 2\exp\left(-\frac{\Pi}{4(L+\epsilon+1)}\right) \le 2Z_{L+\epsilon}\exp\left(-\frac{\Pi}{4(L+\epsilon+1)}\right) \le \frac{\delta}{3};$$

where the last inequality comes from the choice of $\Pi$. □

**Lemma 16** ([28], Lem. B.5). Let $\pi$ be a proper policy such that for some $d > 0$, $V_\pi(s) \le d$ for every non-goal state $s$. Then the probability that the cumulative cost of $\pi$ to reach the goal state from any state $s$ is more than $m$, is at most $2e^{-m/(4d)}$ for all $m \ge 0$. Note that a cost of at most $m$ implies that the number of steps is at most $m/c_{min}$.

## D.7 Putting Everything Together: Sample Complexity Bound

The sample complexity of the algorithm is solely induced by the sample collection procedure (step ?). Recall that we denote by $K$ the index of the round at which the algorithm terminates. With probability at least $1-\frac{2\delta}{3}$, Lem. 13 holds, and so does the event. Hence the algorithm discovers a set of states $\mathcal{K}_K \supseteq S^?_L$. Moreover, from Lem. 14, the algorithm outputs for each $s \in \mathcal{K}_K$ a policy $\pi_s$ with $E[\tau_{\pi_s}(s_0 \to s)] \le V^?_{S^?_L}(s) + \epsilon$. Hence we also have $|\mathcal{K}_K| \le S_{L+\epsilon} := |S^?_{L+\epsilon}|$.

We denote by $Z_K := |\mathcal{K}_K| A \Phi(\mathcal{K}_K)$ the total number of samples that the learner needs to collect. From Lem. 15, with probability at least $1-\frac{\delta}{3}$, the total sample complexity of the algorithm is at most

$$\Pi Z_K, \text{ where } \Pi := 4(L+\epsilon+1)\log\frac{6Z_{L+\epsilon}}{\delta}.$$

Now, from Eq. 19 there exists an absolute constant $c > 0$ such that DisCo selects as allocation function

$$\phi: X \mapsto c\left(\frac{L^4 \Gamma(X)}{\epsilon^2}\log^2\frac{LSA}{\epsilon} + \frac{L^2 |X|}{\epsilon}\log\frac{LSA}{\epsilon}\right);$$

where

$$\Gamma(X) := \max_{(s,a)\in X\times A}\left(\sum_{s^0\in X}\sqrt{p(s^0|s;a)(1-p(s^0|s;a))}\right)^2:$$

The total requirement is $\phi(\mathcal{K}_K)$. Note that from Cauchy-Schwarz's inequality, we have

$$\phi(\mathcal{K}_K) \le \Gamma_K := \max_{(s,a)\in\mathcal{K}_K\times A}\|\{p(s^0|s;a)\}_{s^0\in\mathcal{K}_K}\|_0 \, |\mathcal{K}_K|:$$

Combining everything yields with probability at least $1 - \delta$,

$$Z_K = \tilde{O}\left(\frac{L^5 |\mathcal{K}_K| |K_K| A}{\varepsilon^2} + \frac{L^3 |K_K|^2 A}{\varepsilon}\right).$$

We finally use that $|K_K| \leq S_{L+\varepsilon}^\downarrow$ from Lem. 14, which implies that

$$C_{AX^?}(\text{DisCo}, L, \varepsilon, \delta) = \tilde{O}\left(\frac{L^5 \Gamma_{L+\varepsilon} S_{L+\varepsilon} A}{\varepsilon^2} + \frac{L^3 S_{L+\varepsilon}^2 A}{\varepsilon}\right),$$

where $\Gamma_{L+\varepsilon} := \max_{(s,a) \in S_{L+\varepsilon}^\downarrow \times A} |\{s' | p(s'|s, a) > 0\}_{s' \in S_{L+\varepsilon}^\downarrow}| \geq 0$. This concludes the proof of Thm. 1.

## D.8 Proof of Corollary 1

The result given in Cor. 1 comes from retracing the analysis of Lem. 14 and therefore Lem. 10 by considering non-uniform costs between $[c_{min}, 1]$ instead of costs all equal to 1. Specifically, Eq. 20 needs to account for the inverse dependency on $c_{min}$ of the simulation lemma of Lem. 3. This induces the final $\varepsilon' = c_{min} \varepsilon$ accuracy level achieved by the policies output by DisCo. There remains to guarantee that condition 8 of Lem. 3 is verified. In particular the condition holds if $\eta (L + 1 + \varepsilon') \leq 2 c_{min}$, where $\eta$ is the model accuracy prescribed in the proof of Lem. 10. We see that this is the case whenever we have $\varepsilon = O(L c_{min})$ due to the fact that $\eta = \Theta(\varepsilon / L^2)$.

## D.9 Computational Complexity of DisCo

The overall computational complexity of DisCo can be expressed as $\sum_{k=1}^{K} |W_k| \cdot C(\text{OVI}_{SSP})$, where $C(\text{OVI}_{SSP})$ denotes the complexity of an $\text{OVI}_{SSP}$ procedure and where we recall that $K$ denotes the (random) index of the last round during which the algorithm terminates. Note that it holds with high probability that $K \leq |S_{L+\varepsilon}^\downarrow|$ and $|W_k| \leq 2LA |K_k| \leq 2LA |S_{L+\varepsilon}^\downarrow|$. Moreover $C(\text{OVI}_{SSP})$ captures the complexity of the value iteration (VI) algorithm for SSP, which was proved in [34] to converge in time quadratic w.r.t. the size of the considered state space ($|K_k|$) and $\|V^\star\|_1 / c_{min}$. Here we have $c_{min} = 1$, and we can easily prove that in all the SSP instances considered by DisCo, the optimal value function $V^\star$ verifies $\|V^\star\|_1 = O(L^2)$, due to the restriction of the goal state to $W_k$ (indeed this restriction implies that there exists a state-action pair in $K_k \times A$ that transitions to the goal state with probability $(1 - \varepsilon / L)$ in the true MDP). Putting everything together gives DisCo's computational complexity. Interestingly, we notice that while it depends polynomially on $S_{L+\varepsilon}$, $L$ and $A$, it is independent from $S$, the size of the global state space.

# E The UcbExplore Algorithm [1]

## E.1 Outline of the Algorithm

The UcbExplore algorithm was introduced by Lim and Auer [1] to specifically tackle condition $AX_L$. The algorithm maintains a set $K$ of "controllable" states and a set $U$ of "uncontrollable" states. It alternates between two phases: state discovery and policy evaluation. In a state discovery phase, new candidate states are discovered as potential members of the set of controllable states. Any policy evaluation phase is called a round and it relies on an optimistic principle: it attempts to reach an "optimistic" state $s$ (i.e., the easiest state to reach based on information collected so far) among all the candidate states by executing an optimistic policy $\pi_s$ that minimizes the optimistic expected hitting time truncated at a horizon of $H_{UCB} := \frac{1}{\varepsilon}(L + L^2 \varepsilon)^{-1}e$. Within the round of evaluation of policy $\pi_s$, the algorithm proceeds through at most $t_{UCB} := \frac{1}{\varepsilon}6L^3 \varepsilon^{-3} \log\left(\frac{16|K|^2}{\delta}\right)^{-1}$ episodes, each of which begins at $s_0$ and ends either when $\pi_s$ successfully reaches $s$ or when $H_{UCB}$ steps have been executed. If the empirical performance of $\pi_s$ is poor (measured through a performance check done after each episode), the round is said to have failed. Otherwise, the round is successful which means that $s$ is controllable and an acceptable policy ($\pi_s$) has been discovered. A failure round leads to selecting another candidate state-policy pair for evaluation, while a success round leads to a state discovery phase which in turn adds more candidate states for the subsequent rounds. As explained in App. A, UcbExplore is unable to tackle the more challenging objective $AX^?$.

### E.2 Minor Issue and Fix in the Analysis of UcbExplore

The key insight of UcbExplore is to bound the number of failure rounds of the algorithm, by lower- and upper-bounding the so-called "regret" contribution of failure rounds, where the regret of a failure round $k$ is defined as

$$\sum_{j=1}^{e_k} \left[ H_{UCB} \, L - \sum_{i=0}^{1} r_i \right];$$

where $e_k$ $_{UCB}$ is the actual number of episodes executed in round $k$ and where the reward $r_i \in \{0, 1\}$ is equal to 1 only if the state is the goal state. However, upper bounding the regret contribution of failure rounds implies applying a concentration inequality only on specific rounds, that are chosen given their empirical performance. Hence Lim and Auer [1, Lem. 18] improperly use a martingale argument to bound a sum whose summands are chosen in a non-martingale way, i.e., depending on their realization.

To avoid the aforementioned issue, one must upper and lower bound the cumulative regret of the entire set of rounds and not only the failure rounds in order to obtain a bound on the number of failure rounds. However, this would yield a sample complexity that has a second term scaling as $\Theta(1/\varepsilon^4)$. Following personal communication with the authors, the fix is to change the definition of regret of a round, making it equal to

$$\sum_{j=1}^{e_k} \left[ \theta_{H_{UCB}}(s_0 \to s) - \sum_{i=0}^{H_{UCB}-1} r_i \right];$$

where $s$ is the considered goal state and $\theta_{H_{UCB}}(s_0 \to s)$ is the optimistic $H_{UCB}$-step reward (where the reward is equal to 1 only at state $s$). With this new definition, it is possible to recover the sample complexity provided in [1] scaling as $\Theta(\varepsilon^{-3})$.

### E.3 Issue with a Possibly Infinite State Space

Lim and Auer [1] claim that their setting can cope with a countable, possibly infinite state space. However, this leads to a technical issue, which has been acknowledged by the authors via personal communication and as of now has not been resolved. Indeed, it occurs when a union bound over the unknown set $U$ is taken to guarantee high-probability statements (e.g., the Lem. 14 or 17 of [1]). Yet for each realization of the algorithm, we do not know what the set $U$, or equivalently $K$, looks like, hence it is improper to perform a union bound over a set of unknown identity. Simple workarounds to circumvent this issue are to impose a finite state space, or to assume prior knowledge over a finite superset of $U$. In this paper we opt for the first option. It remains an open and highly non-trivial question as to how (and whether) the framework can cope with an infinite state space.

### E.4 Effective Horizon of the AX Problem and its Dependency on $L$

UcbExplore [1] designs finite-horizon problems with horizon $H_{UCB} := \lceil L + L^2 \varepsilon^{-1} \rceil$ and outputs policies that reset every $H_{UCB}$ time steps. In the following we prove that the effective horizon of the AX problem actually scales as $L \log(L \varepsilon^{-1})$, i.e., only logarithmically w.r.t. $\varepsilon^{-1}$. We begin by defining the concept of "resetting" policies as follows.

**Definition 11.** For any $\pi \in \Pi$ and horizon $H \geq 0$, we denote by $\pi^H$ the non-stationary policy that executes the actions prescribed by $\pi$ and performs the RESET action every $H$ steps, i.e.,

$$\pi_t^H(a|s) := \begin{cases} \text{RESET} & \text{if } t \equiv 0 \ (\mathrm{mod}\, H); \\ \pi(a|s) & \text{otherwise.} \end{cases}$$

We denote by $\Pi^H$ the set of such "resetting" policies.

The following lemma captures the effective horizon $H_{eff}$ of the problem, in the sense that restricting our attention to $\pi^H(S_L^{\to})$ for $H \geq H_{eff}$ does not compromise the possibility of finding policies that achieve the performance required by $AX^?$ (and thus also by $AX_L$).

Lemma 17. For any $\varepsilon \in (0,1]$ and $L \geq 1$, whenever

$$H \geq H_{\text{eff}} := 4(L+1)\,\log\!\left(\frac{4(L+1)}{\varepsilon}\right);$$

we have for any $s^g \in S_L^!$,

$$\min_{\pi_H \in \Pi_H(S_L^!)} v_{\pi_H}(s_0 \to s^g) \leq V_{S_L^!}^?(s_0 \to s^g) + \varepsilon.$$

Proof. Consider any goal state $s^g \in S_L^!$. Set $\varepsilon^0 := \frac{\varepsilon}{2(L+1)} \leq \frac{1}{2}$. Denote by $\pi \in \Pi(S_L^!)$ the minimizer of $V_{S_L^!}^?(s_0 \to s^g)$. For any horizon $H \geq 0$, we introduce the truncated value function $v_{\pi;H}(s \to s^g) := E[\tau(s \to s^g) \wedge H]$ and the tail probability $q_{\pi;H}(s \to s^g) := P(\tau(s \to s^g) > H)$. Due to the presence of the RESET action, the value function of $\pi$ can be bounded for all states $s \in S_L^! \cap \{s^g\}$ as

$$v_\pi(s \to s^g) \leq V_{S_L^!}^?(s_0 \to s^g) + 1 \leq L + 1:$$

This entails that the probability of the goal-reaching time decays exponentially. More specifically, we have

$$q_{\pi;H}(s_0 \to s^g) \leq 2\exp\!\left(-\frac{H}{4(L+1)}\right) \leq \varepsilon^0, \tag{21}$$

where the first inequality stems from Lem. 16 and the second inequality comes from the choice of $H \geq 4(L+1)\,\log\frac{2}{\varepsilon^0}$. Furthermore, we have $\tau(s \to s^g) \wedge H \leq \tau(s \to s^g)$ and thus $E[\tau(s \to s^g) \wedge H] \leq E[\tau(s \to s^g)]$. Consequently,

$$v_{\pi;H}(s_0 \to s^g) \leq v_\pi(s_0 \to s^g) = V_{S_L^!}^?(s_0 \to s^g): \tag{22}$$

Now, from [1, Eq. 4], the value function of $\pi$ can be related to its truncated value function and tail probability as follows

$$v_{\pi_H} = \frac{v_{\pi;H} + q_{\pi;H}}{1 - q_{\pi;H}}: \tag{23}$$

Plugging Eq. 21 and 22 into Eq. 23 yields

$$v_{\pi_H}(s_0 \to s^g) \leq \frac{V_{S_L^!}^?(s_0 \to s^g) + \varepsilon^0}{1 - \varepsilon^0}:$$

Notice that the inequalities $\frac{1}{1-x} \leq 1 + 2x$ and $\frac{x}{1-x} \leq 2x$ hold for any $0 < x \leq \frac{1}{2}$. Applying them for $x = \varepsilon^0$ yields

$$\frac{V_{S_L^!}^?(s_0 \to s^g) + \varepsilon^0}{1 - \varepsilon^0} \leq (1 + 2\varepsilon^0)V_{S_L^!}^?(s_0 \to s^g) + 2\varepsilon^0:$$

From the inequality $V_{S_L^!}^?(s_0 \to s^g) \leq L$ and the definition of $\varepsilon^0$, we finally obtain

$$v_{\pi_H}(s_0 \to s^g) \leq V_{S_L^!}^?(s_0 \to s^g) + \varepsilon;$$

which completes the proof. □

Lem. 17 reveals that the effective horizon $H_{\text{eff}}$ of the AX problem scales only logarithmically and not linearly in $\varepsilon^{-1}$. This highlights that the design choice of UcbExplore to tackle finite-horizon problems with horizon $H_{\text{UCB}}$ unavoidably leads to a suboptimal dependency in $\varepsilon$ in its AX$_L$ sample complexity bound. In contrast, by designing SSP problems and thus leveraging the intrinsic goal-oriented nature of the problem, DisCo can (implicitly) capture the effective horizon of the problem. This observation is at the heart of the improvement in the $\varepsilon$ dependency from $\tilde{O}(\varepsilon^{-3})$ of UcbExplore [1] to $\tilde{O}(\varepsilon^{-2})$ of DisCo (Thm. 1).

# F Experiments

This section complements the experimental findings partially reported in Sect. 5. We provide details about the algorithmic configurations and the environments as well as additional experiments.

## F.1 Algorithmic Configurations

**Experimental improvements to UcbExplore [1].** We introduce several modifications to UcbExplore in order to boost its practical performance. We remove all the constants and logarithmic terms from the requirement for state discovery and policy evaluation (refer to Fig. 1]). Furthermore, we remove the constants in the definition of the accuracy $\varepsilon' = \varepsilon/L$ used by UcbExplore (while their original algorithm requires $\varepsilon'$ to be divided by $8$, we remove this constant). We also significantly improve the planning phase of UcbExplore [1, Fig. 2]. Their procedure requires to divide the samples into $H := (1 + 1/\varepsilon') L$ disjoint sets to estimate the transition probability of each stage $h$ of the finite-horizon MDP. This substantially reduces the accuracy of the estimated transition probability since for each stage $h$ only $N_k(s, a)/H$ are used. In our experiments, we use all the samples to estimate a stationary MDP ($\hat{p}_k(s'|s, a) = N_k(s, a, s')/N_k(s, a)$) rather than a stage-dependent model. Estimating a stationary model instead of bucketing the data is simpler and more efficient since leads to a higher accuracy of the estimated model. To avoid to move too far away from the original UcbExplore, we decided to define the confidence intervals as if bucketing was used. We thus consider $N_k(s, a) = N_k(s, a)/H$ for the construction of the confidence intervals. For planning, we use the optimistic backward induction procedure as in [30]. We thus leverage empirical Bernstein inequalities —which are much tighter— rather than Hoeffding inequalities as suggested in [1]. In particular, we further approximate the bonus suggested in [30, Alg. 4] as

$$b_h(s, a) = \sqrt{\frac{\mathrm{Var}_{s' \sim \hat{p}_k(\cdot|s,a)}[V_{k;h+1}(s')]}{N_k(s, a) - 1}} + \frac{(H - h)}{N_k(s, a) - 1}.$$

For DisCo, we follow the same approach of removing constants and logarithmic terms. We thus use the definition of $\beta$ as in Thm. 1 with $\delta = 1$ and without log-terms. For planning, we use the procedure described in App. D with $\phi_k(s, a, s') = \sqrt{\frac{\hat{p}_k(s'|s,a)(1-\hat{p}_k(s'|s,a))}{N_k(s,a)-1}} + \frac{1}{N_k(s,a)-1}$. Finally, in the experiments we use a state-action dependent value area $K_k(s, a) = \sum_{s' \notin 2K_k} \sqrt{\hat{p}_k(s'|s,a)(1-\hat{p}_k(s'|s,a))}^2$ instead of taking the maximum over $(s, a)$.

Even though we boosted the practical performance of UcbExplore w.r.t. the original algorithm proposed in [1] (e.g., the use of Bernstein), we believe it makes the comparison between DisCo and UcbExplore as fair as possible.

## F.2 Confusing Chain

The *confusing chain* environment referred to in Sect. 5 is constructed as follows. It is an MDP composed of an initial state $s_0$, a chain of length $C$ (states are denoted by $s_1, \ldots, s_C$) and a set of $K$ confusing states ($s_{C+1}, \ldots, s_{C+K}$). Two actions are available in each state. In state $s_0$ we have a forward action $a_0$ that moves to the chain with probability $p_c$ ($p(s_1|s_0, a_0) = p_c$ and $p(s_0|s_0, a_0) = 1 - p_c$) and a confusing action that has uniform probability of reaching any confusing state ($p(s_i|s_0, a_1) = 1/K$ for any $i \in \{C+1, \ldots, C+K\}$). In the confusing states, all actions move deterministically to the end of the chain ($p(s_C|s_i, a) = 1$ for any $i \in \{C+1, \ldots, C+K\}$ and $a$). In each state of the chain, there is a forward action $a_0$ that behaves as in $s_0$ ($p(s_{\min(C,i+1)}|s_i, a_0) = p_c$ and $p(s_i|s_i, a_0) = 1 - p_c$, for any $i \in \{1, \ldots, C-1\}$) and a skip action $a_1$ that moves $m$ states ahead with probability $p_{skip}$ ($p(s_{\min(C,i+m)}|s_i, a_0) = p_{skip}$ and $p(s_i|s_i, a_0) = 1 - p_{skip}$, for any $i \in \{1, \ldots, C-1\}$). Finally, $p(s_0|s_C, a) = 1$ for any action $a$. In our experiments, we set $m = 4$, $p_{skip} = 1/3$, $p_c = 1$, $C = 5$, $K = 6$, $L = 4.5$.

| $\varepsilon$ | DisCo | UcbExplore-Bernstein |
|---|---|---|
| 0.1 | $374{,}263\ (13{,}906)$ | $5{,}076{,}688\ (92{,}643)$ |
| 0.2 | $105{,}569\ (4{,}645)$ | $636{,}580\ (13{,}716)$ |
| 0.4 | $29{,}160\ (829)$ | $108{,}894\ (2{,}305)$ |
| 0.6 | $15{,}349\ (475)$ | $40{,}538\ (805)$ |
| 0.8 | $9{,}891\ (244)$ | $21{,}270\ (441)$ |

Table 2: Sample complexity of DisCo and UcbExplore-Bernstein, on the confusing chain domain. Values are averaged over 50 runs and the 95%-confidence interval of the mean is reported in parenthesis.

| | UcbExplore-Bernstein | | | | | |
|---|---|---|---|---|---|---|
| $\varepsilon$ | Expected hitting time $v^\star(s_0 \to s_i)$ | | | | | |
| | $s_0$ | $s_1$ | $s_2$ | $s_3$ | $s_4$ | $s_5$ |
| 0.1; 0.2 | 0 | 1 | 2 | 3 | 4 | 4 |
| 0.4 | 0 | 1 | 2 | 3 | 4 | $4.94\ (0.04)$ |
| 0.6 | 0 | 1 | 2 | $3.36\ (0.11)$ | 4 | $4.53\ (0.07)$ |
| 0.8 | 0 | 1 | 2 | $3.38\ (0.11)$ | $4.07\ (0.07)$ | $4.53\ (0.06)$ |

Table 3: Expected hitting time of states $s_i$ of the goal-oriented policy recovered by UcbExplore-Bernstein, on the confusing chain domain. DisCo recovers the optimal goal-oriented policy in all the runs and for all $\varepsilon$. The advantage of DisCo lies in its final policy consolidation step. Values are averaged over 50 runs and the 95%-confidence interval of the mean is reported in parenthesis (it is omitted when equal to 0). This shows that UcbExplore recovers the optimal goal-oriented policy in every run only for $\varepsilon$ equal to 0.1 and 0.2.

**Sample complexity.** We provide in Tab. 2 the sample complexity of the algorithms for varying values of $\varepsilon$. As mentioned in Sect. 5, DisCo outperforms UcbExplore for any value of $\varepsilon$, and increasingly so when $\varepsilon$ decreases. Fig. 7 complements Fig. 2 for additional values of $\varepsilon$.

**Quality of goal-reaching policies.** We now investigate the quality of the policies recovered by DisCo and UcbExplore. In particular, we show that DisCo is able to find the incrementally near-optimal shortest-path policies to any goal state, while UcbExplore may only recover sub-optimal policies. On the confusing chain domain, the intuition is that the set of confusing states $s_D$ makes $s_D$ reachable in just 2 steps but the confusing states are not in the controllable set and thus the algorithms are not able to recover the shortest-path policy to $s_D$. On the other hand, state $s_C$ is controllable through two policies: 1) the policy $\pi_1$ that takes always the forward action $a_0$ reaches $s_C$ in 5 steps; 2) the policy $\pi_2$ that takes the skip action $a_1$ in $s_1$ reaches $s_C$ in 4 steps. We observed empirically that DisCo always recovers policy $\pi_1$ (i.e., the fastest policy) while UcbExplore selects policy $\pi_2$ in several cases. This is highlighted in Tab. 3 where we report the expected hitting time of the policies recovered by the algorithms. This finding is not surprising since, as we explain in Sect. 4 and App. A, UcbExplore is designed to find policies reaching states at most $L$ steps on average, yet it is not able to recover incrementally near-optimal shortest-path policies, as opposed to DisCo.

## F.3 Combination Lock

We consider the combination lock problem introduced in [31]. The domain is a stochastic chain with $S = 6$ states and $A = 2$ actions. In each state $s_k$, action right ($a_1$) is deterministic and leads to state $s_{k+1}$, while action left ($a_0$) moves to a state $s_{k-l}$ with probability proportional to $1/(k-l)$ (i.e., inversely proportional to the distance of the states). Formally, we have that

$$n(x_k; x_l) = \begin{cases} \dfrac{1}{k-l} & \text{if } l < k \\ 0 & \text{otherwise} \end{cases} \quad \text{and} \quad p(x_l \mid x_k; a_0) = \dfrac{n(x_k; x_l)}{\sum_s n(x_k; s)}, \quad j < k$$

We set the initial state to be at $\tfrac{1}{2}$ of the chain, i.e., $2N=3$. The actions in the end states are absorbing, i.e. $p(s_0 \mid s_0; a_0) = 1$ and $p(s_{N-1} \mid s_{N-1}; a_1) = 1$, while the remaining actions behave normally. See Fig. 5 for an illustration of the domain.

Figure 5: Combination lock domain with $S = 6$ states. Expected hitting times from the initial state $s_3$ are $v^{\pi}(s_3 \to s) = (2.18; 1.91; 1.64; 0; 1; 2)$. Consider $L = 3$, the set of incrementally $L$-controllable states is $S^{\dagger}_L = \{s_2; s_3; s_4; s_5\}$. The goal-oriented policy to reach $s_4$ and $s_5$ takes always the right action $a_1$, while the policy for $s_2$ always selects the left action $a_0$.

Figure 6: Proportion of the incrementally $L$-controllable states identified by DisCo and UcbExplore in the combination lock domain for $L = 2.7$ and $\varepsilon = 0.2$. Values are averaged over $20$ runs.

Sample complexity. We evaluate the two algorithms DisCo and UcbExplore on the combination lock domain, for $\varepsilon = 0.2$ and $L = 2.7$. We further boost the empirical performance of UcbExplore by using $N$ instead of $\underline{N}$ for the construction of the confidence intervals (i.e., we do not account for the data bucketing in [1], see App. F.1). To preserve the robustness of the algorithm, we use $\log(|K_k|^2)/(\varepsilon^0)^3$ episodes for UcbExplore's policy evaluation phase (indeed we noticed that the removal of the logarithmic term here sometimes leads UcbExplore to miss some states in $S^{\dagger}_L$ in this domain). For the same reason, in DisCo we use the value $\phi(K_k) = \max_{s;a} \phi(s; a; K_k)$ prescribed by the theoretical algorithm instead of the state-action dependent values used in the previous experiment. We average the experiments over $20$ runs and obtain a sample complexity of $30, 117 (2; 087)$ for DisCo and $90, 232 (2; 592)$ for UcbExplore. Fig. 6 reports the proportion of incrementally $L$-controllable states identified by the algorithms as a function of time. We notice that once again DisCo clearly outperforms UcbExplore.

Figure 7: Proportion of the incrementally $L$-controllable states identified by `DisCo` and `UcbExplore` on the confusing chain domain for $L = 4.5$ and $\varepsilon \in \{0.1, 0.2, 0.4, 0.6, 0.8\}$. Values are averaged over 50 runs. `UcbExplore` uses Bernstein confidence intervals for planning.