[Reviews · NeurIPS 2020]

Review 1

Summary and Contributions: The paper proposes an algorithm DISCO to compute the \epsilon-optimal goal-conditioned policies covering states which are L-incrementally reachable from a reference state. It improves over the previous sample complexity bound of UCBEXPLORE in terms of L and epsilon but worsens in terms of S. DISCO is further adopted to find epsilon/c-optimal policy for cost-sensitive SSP.

Strengths: 1. The paper proposes a new algorithm which is well explained and modular. 2. The paper provides a new bound on sample complexity while satisfying the stronger AX* condition. 3. The paper draws parallels and contrasts to existing work [1] in detail which is very helpful. 4. Disco also provides guarantees for finding optimal policies in cost-sensitive SSP. 5. This paper can be a good step to analyse popular deep-RL methods like Go-Explore.

Weaknesses: 1. Not enough comparison with the recent reward-free exploration work [23]. 2. Discussing non-theoretical motivation between the L-controllable states and more importantly its limitations. 3. There is no analysis or discussion of computational complexity which seems to be pretty expensive. 4. Error in empirical evidence section is a bit misleading.

Correctness: 1. I don't see any error in theoretical proofs or claims. 2. The erratum of App G.1. should be shifted to main text.

Clarity: It is compact due to the theoretical flavour but clearly written.

Relation to Prior Work: Relation to prior work is discussed in detail. One remaining loophole is discussing the contrast in strategy with that of [23]. The idea of visiting the "good" states while not "wasting" time on "far" states is very close to the idea of \delta-significant states in [23]. Discussing similarity and contrast in the paper would be helpful to posit the work.

Reproducibility: Yes

Additional Feedback: 1. Formatting: a. The labels in Fig 2 are almost unreadable. Writing "proportion of" instead of "\%" is suggested. b. "reachable states" is used in intro while "controllable" in next ones. Homogenising terminology will be helpful. c. In line 17 of Algo 1, Use "K_final" or something else than "K_k" for unambiguity. 2. Typos: a. Page 8 line 284: "policy it" -> "policy which/that" b. Page 8 last para: using caps and "." is writing is awkward. It's more of a ppt style than grammatical. 3. Discussing the computational complexity of the algorithm would be crucial, specially that of OVI_SSP at different phases of algo 1. 4. Changes on the points mentioned in correctness and relation to prior work are crucial. 5. What are the limitations of this setup or benefits in comparison with episodic, communicating, and non-communicating MDPs? 6. How the values of L can change or can be set? For example, for a communicating MDP, one can imagine it to be upper bounded by D. Some explaination in this line would be helpful as presently it seems like an artificial construction/hyperparameter. 7. How does setting epsilon to upper limit 1 (as set in algo) affects the selection of W_k? Though making it small is good from policy point of view, it can be also thought as a slack in L and in reality it can go up to 2. After authors' response: The authors' response seems quite promising and clarifying. It would be helpful to incorporate the comments and discussions of the response in the main paper. Specially, the discussions on "Motivation/limitations of the incremental framework", "DISCO for cost-sensitive tasks", "Computational complexity", "Comparison with reward-free exploration" should be added in the final version.


Review 2

Summary and Contributions: In this paper, the authors are interested in optimizing the exploration of the state space in MDPs, when, for example, the reward function is sparse. The aim is to find out which states can be reached in a defined average number of time steps, and to compute the associated policy. This work builds on the incremental exploration setting of Lim et al. 2012 that proposed UcbExplore. After formalizing the framework and notions of state controllability, the authors propose a new criterion for policy optimization that is stronger than the previous one. They then present their model-based algorithm DisCo which returns (incrementally) controllable states, and associated policies. They deduce a bound over the sample complexity that is often better than the one of UCBEXPLORE. Finally, the authors present an empirical evaluation of their algorithm and conclude by discussing the links with deep-RL methods.

Strengths: This paper is very interesting, well written and well presented. It focuses on autonomous exploration, which is a problem of interest in reinforcement learning, especially when the reward function is sparse. This paper is therefore very relevant to the NeurIPS community.

Weaknesses: My main concerns with this paper are the lack of sketches of proof in the main paper for the theoretical results and the few experimental results. Indeed, the main contributions are Theorem 1 and Corollary 1, so it would be relevant to provide short justifications for them. Finally, the algorithm is only tested on a single environment and compared to UCBEXPLORE for a single value of L (the average number of steps to reach the desired states).

Correctness: I didn't read any statements that I thought were wrong.

Clarity: The paper is well written. I just have some remarks: line 128 < all L-controllable states > all incrementally L-controllable states definition 6: It could be clearer to write that s' is the unique goal of the SSP M_k' Figure 2: It might be clearer to show the epsilon value on the figures. line 273: The sentence "For both the algorithms we remove the logarithmic terms and all the constants." is not clear enough.

Relation to Prior Work: The authors highlighted their contribution well by comparing their work with UcbExplore and other really recent works.

Reproducibility: No

Additional Feedback:


Review 3

Summary and Contributions: The paper introduces a new algorithm for efficiently exploring states that are within an expected L steps from an initial state s_0 and solving the all-pairs shortest path problem between those states. It improves over the UCBExplore algorithm that focused just on the first of those problems with a hard horizon of L rather than an expectation. The new algorithm proceeds by incrementally growing a set of known states in each round, sampling new actions from all of these to either refine the existing shortest path calculations or discover unknown states that are feasible, targeting the closest of these for exploration in the next round. Theoretical bounds on the sample complexity are proven and compare favorably to UCBExplore and a numerical simulation shows the performance is translated to empirical gains.

Strengths: The Disco algorithm is very clever, building on the long history of algorithms growing a set of known states but also introducing new machinery in the calculations and adapting them to the 2-objective problem in this paper. The pseudocode is well laid out, the text description is very intuitive, and the supplemental material was helpful understanding the nuances of how a subcomponent like OptiVI was used. The theory is novel and compares favorably with UCBExplore, though I think some more caveats may be needed (see below). The empirical results are on a single domain but useful here just to show that the theory translates to a real performance gain.

Weaknesses: Overall I am happy with the paper but I think there are some areas where wording can be cleaned up or where claims need to be given with more context or nuance, particularly around the comparison to UCBExplore bounds and the discussion of the algorithm for general RL beyond stochastic shortest path problems, as outlined in the detailed sections below.

Correctness: In the comparison of the bounds to UCBExplore’s bounds starting at line 212, there are a few claims that seem like they need more evidence or at least a numerical example. Specifically, the condition on line 224 that says when S <= L^3/epsilon^2 the bound of Disco is preferable is hard to understand because the terms are not particularly relatable. S,L, and epsilon are all fairly independent. It would be good to see some numbers here even from the existing experiments saying how common this case is. Would we expect RL scenarios to have this property? Or just some? Similarly, on line 228 it is implied that epsilon is the dominant term in the algorithm but only intuition is given here. Are there numerical results to back this up? Usually in MDP learning S and \Gamma (branching) are much more important terms than epsilon.

Clarity: The paper is generally very well written, I found it very enlightening and the flow is terrific. The footnotes are helpful, the supplementary material fills in gaps but is not necessary (so it’s in the right place) and the algorithm is very intuitively laid out. My only writing suggestion is to add some of the motivational material from lines 126-139 to the intro to help set the problem concretely in the early going as I wasn’t sure where the paper was going on the first page or so. Typos and minor notes: Line 33: lack of substantial -> lack substantial Figure 1- It should be pointed out in the discussion of these examples that the definitions of L-controllable states, etc. are computed without resets, since the reset action actually gives you a way of returning to those states enough to control them. Line 174 – The word “pruning” creates confusion here because you are not really pruning states forever (e.g. never considering them again), you’re pruning the candidate set for the next round, which is technically what you said but the wording would be better as “restricting the candidate set in each round” since “pruning” usually indicates an irreversible elimination.

Relation to Prior Work: The work is well placed in the literature but I had one issue with the comparison to UCBExplore -- the paper commonly refers to UCBExplore as restricted to finite horizon problems and the current paper being free of this constraint, which is technically true. But Disco is restricted to an expected horizon based on L (just not a hard one), so it is better described as a “relaxed horizon constraint” or some other term.

Reproducibility: Yes

Additional Feedback: UPDATE: I thank the authors for their response. Their points on the motivation are well taken and should be incorporated into the final version. I agree with the clarification they propose and also the MDP family investigation sounds like good future work. The section on cost sensitive stochastic shortest path guarantees on line 241 seems to overstate Disco’s capabilities in a general MDP. In fact, throughout the paper Disco is often referred to in the context of general MDP RL but really it is restricted to SSP RL. In this section, the claim that Disco “could learn to tackle any RL task faster” (line 248) is not supported. Disco may be used to find a reachable set of states and another RL algorithm that has to learn about reward could be used , but there is no evidence that would be faster in the general RL setting. Similarly, in Corollary 1, the claim that Disco can compute general RL policies given a cost function seems too far – value iteration can compute that given the results from Disco, but there I don’t see any new machinery in Disco that would be used for this purpose. Mentioning these cases is fine, but I think the wording needs to be pared back a little.


Review 4

Summary and Contributions: The paper proposes an algorithm for pure exploration in MDPs. It gives an asymptotic performance guarantee, which shows performance is better in terms of the maximum navigation time (L) and the tolerance epsilon compared to an existing guarantee for UCBexplore. Update after reviewer discussion: thanks for clarifying my points!

Strengths: A. Pure exploration in MDPs in a very relevant problem. In many practical cases, we either want to to solve several versions of an MDP with different rewards but the same transition kernels or else the reward is much more expensive to obtain than access to simulated transitions. B. I agree with the authors that the performance measure AX* is more natural than AXL - we do pure exploration to get efficient policies, not policies that get us to the goal eventually. C. The paper is VERY well-written.

Weaknesses: This work does not have any major weaknesses. I list a couple of minor points below. A. Performance in the table-lookup setting The immediate practical applicability of the paper remains limited. In small tabular MDPs, where this work can be applied directly, the performance difference shown in Figure 2 isn't huge (although I respect the authors for putting the figure in - honest evaluation of claims makes the paper stronger). B. Practical applicability in large MDPs In practical cases where we have MDPs with huge or infinite state spaces we would have to construct some kind of a "latent MDP" to profitably apply the algorithm. Constructing this "latent MDP" well is probably a harder problem than pure exploration in a tabular MDP. I know that covering this use case would be very hard in a single paper. However, you do mention Montezuma's Revenge in the intro so I was kind of expecting a discussion along these lines. Having said that, what the algorithm does, it does really well, so I don't think this is a major problem in terms of the score. C. How tight are the bounds? Theorem 1 is stated in the tilde-O notation, which ignores constants a,b and lower-order terms. Is it possible to evaluate the bounds in practice? It would be useful to discuss how far the derived bounds are from actual performance on the example given in Section 5.

Correctness: I have not found any problems with the correctness of results shown in the paper.

Clarity: The exposition is exemplary. The paper is very well-written and clear. Minor requests: - on page 3, is "L-reachability" and "L-controllability" the same thing (I think so, but wanted to check)? - in line 1013 (Appendix), you say exactly what it means for the algorithm to converge. Can you denote convergence point with vertical lines in Figure 2?

Relation to Prior Work: I am not aware of relevant prior work that wasn't mentioned.

Reproducibility: Yes

Additional Feedback: Keep up the good work!

[Author Response · NeurIPS 2020]

**We thank the reviewers for their comments and insightful reviews.** We will integrate all the useful suggestions in
the revised version of the paper.

**[R1] Comparison with reward-free exploration [23].** On a high level, both approaches build accurate estimates
of the transitions on a specific (unknown) state space of interest: "significant" states within $H$ steps for [23] and
incrementally $L$-controllable states $S_L^{\rightarrow}$ for DISCO. While the two concepts are somewhat related ($H \equiv L$ are the
horizons of interest), [23] focuses on finite-horizon problems and we consider the more general goal-conditioned setting.
Resetting after every $L$ steps (as in finite-horizon) would not allow identifying the states in $S_L^{\rightarrow}$. This explains the
distinct technical tools used: while [23] deploys finite-horizon no-regret algorithms, DISCO leverages SSP tools. The
bound-wise comparison is also interesting. While $\epsilon$, $A$ and $H \equiv L$ dependencies match, [23]'s dependency on the
global state space $S$ is polynomial, whereas DISCO's is only *logarithmic* as the main dependency is w.r.t. $|S_{L+\epsilon}^{\rightarrow}|$. This
shows that DISCO effectively adapts to the state space of interest and it ignores all other states.

**[R1] Computational complexity.** The overall complexity can be expressed as $\sum_{k=1}^{K} |\mathcal{W}_k| \cdot C(\text{OVI}_{\text{SSP}})$, with
$C(\text{OVI}_{\text{SSP}})$ the complexity of an $\text{OVI}_{\text{SSP}}$ procedure. Note that $K \leq |S_{L+\epsilon}^{\rightarrow}|$ and $|W_k| \leq 2LA|\mathcal{K}_k| \leq 2LA|S_{L+\epsilon}^{\rightarrow}|$. The
VI algorithm for SSP was proved in [37] to converge in time quadratic w.r.t. the size of the considered state space
(here, $\mathcal{K}_k$) and $\|V^\star\|_\infty / c_{\min}$. Here $c_{\min} = 1$, and we can prove that in all SSPs considered by DISCO, the optimal
value function $V^\star$ verifies $\|V^\star\|_\infty = O(L^2)$ due to the restriction of the goal in $\mathcal{W}_k$. Putting everything together gives
DISCO's complexity. Interestingly, it only depends on $|S_{L+\epsilon}^{\rightarrow}|$ and is independent from the global state space size $S$.

**[R1, R3] Motivation/limitations of the incremental framework.** We believe this setting effectively captures the
intuition that an agent progressively expands its knowledge of the environment by leveraging closer well-controlled
states to achieve further states that are more difficult to reach. Interestingly, recent goal-conditioned algorithms
for unsupervised RL or learning with sparse reward (see e.g., [21,22,33]) make the *implicit* assumption that the
environment's states satisfy the incremental controllability condition of Def. 4, in the sense that they strive to train a
policy to reach closer states before moving forward in exploring and controlling other states. Nonetheless, the definition
of $S_L^{\rightarrow}$ may be too restrictive as it excludes states that are $L$-controllable but may require passing through states that are
not. While considering all $L$-controllable states in $\mathcal{S}_L$ would inevitably hit the impossibility result proved by [1], we
believe it is possible to relax the strict incrementality condition of $S_L^{\rightarrow}$ without affecting the learnability of the problem.

**[R1] On $L$.** In DISCO we can gradually increase the value of $L$ without restarting the algorithm from scratch, unlike in
UcbExplore. This allows tuning the parameter online according to the desired behavior. In particular, in the case of
communicating MDPs, one may perform a sort of doubling trick: $L = 2, 4, 8, \ldots, 2^n$, where the unknown $n$ satisfies
$2^{n-1} \leq D \leq 2^n$. Once $2^n$ the algorithm would indeed discover all states in the MDP and we can stop it. Crucially, the
total sample complexity would be (up to logarithmic factors) the same of DISCO run with the final value of $L$.

**[R1] Upper limit on $\epsilon$.** We had set $\epsilon \leq 1$ for ease of analysis, as assumed in e.g., [35]. If it may larger ($\epsilon \leq \epsilon_{\max}$), then
the definition of $\mathcal{W}_k$ would indeed have to be modified accordingly (replacing $1 - \epsilon/2$ by $1 - \epsilon/(2\epsilon_{\max})$).

**[R2] Proof sketch.** A sketch of the proof of Thm. 1 is currently available in App. B. In case of acceptance we will use
the extra page to bring it to the main text so that it indeed contains proof intuition (likewise for Cor. 1).

**[R2] Additional experiments.** Since [1] did not report any numerical study of UcbExplore, in our paper we focused on
two simple environments where it is still relatively easy to interpret the behavior of the algorithms and their performance.
We will include additional experiments for varying $L$ in the final version.

**[R3] Bound dependencies, comparison.** In the condition on line 224, the number of states $S_{L+\epsilon} := |S_{L+\epsilon}^{\rightarrow}|$ directly
depends on $L$ and $\epsilon$ (more precisely it increases with both). As such, all parameters are connected to each other and it
may not be trivial to determine values for which the condition holds. In the environments considered in our experiments,
the condition holds for the chosen values of $\epsilon$ and $L$. We agree an interesting direction for future investigation is to
identify *families* of MDPs where $S_{L+\epsilon}$ is an explicit function of $L$ (e.g., constant, linear, polynomial, exponential).

**[R3] DISCO for cost-sensitive tasks.** We agree the current discussion is poorly phrased. DISCO indeed does not
perform any additional learning. In fact, DISCO returns an estimated model (on which OVI is run) that is sufficiently
accurate w.r.t. the true model restricted to $S_L^{\rightarrow}$. This property guarantees that the SSP policy returned by OVI is
near-optimal for any cost function. Interestingly, it may also be used to compute accurate policies for e.g., finite-horizon
RL tasks restricted on $S_L^{\rightarrow}$ (by leveraging the simulation lemma of Lem. 8). We will clarify this part.

**[R4] Complete bound.** We can retrace the exact terms from the analysis to provide a sample complexity bound
with constants and logs. This will indeed allow to evaluate the bound w.r.t. the performance. We note that while
the performance is partially tied to the bound via the choice of allocation $\phi$, samples are in practice shared between
sample-collection attempts, and in addition the $O(L)$ cost to collect each sample is often loose for states close to $s_0$.

**[R4]** "$L$-reachability" and "$L$-controllability" are indeed the same concept, we will unify the terminology.

[Meta-Review · NeurIPS 2020]

Overall the reviewers agreed that the paper makes a strong contribution.